# Advancing diagnostic performance and clinical usability of neural networks via adversarial training and dual batch normalization

Tianyu Han[1 ✉], Sven Nebelung[2], Federico Pedersoli[2], Markus Zimmermann[2], Maximilian Schulze-Hagen[2], Michael Ho[3], Christoph Haarburger[3], Fabian Kiessling [4,5,6], Christiane Kuhl[2], Volkmar Schulz [1,5,6,7 ✉] & Daniel Truhn[2,7 ✉]

Unmasking the decision making process of machine learning models is essential for implementing diagnostic support systems in clinical practice. Here, we demonstrate that adversarially trained models can significantly enhance the usability of pathology detection as compared to their standard counterparts. We let six experienced radiologists rate the interpretability of saliency maps in datasets of X-rays, computed tomography, and magnetic resonance imaging scans. Significant improvements are found for our adversarial models, which are further improved by the application of dual-batch normalization. Contrary to previous research on adversarially trained models, we find that accuracy of such models is equal to standard models, when sufficiently large datasets and dual batch norm training are used. To ensure transferability, we additionally validate our results on an external test set of 22,433 X-rays. These findings elucidate that different paths for adversarial and real images are needed during training to achieve state of the art results with superior clinical interpretability.

[1] Physics of Molecular Imaging Systems, Experimental Molecular Imaging, RWTH Aachen University, Aachen, Germany. [2] Department of Diagnostic and Interventional Radiology, University Hospital Aachen, Aachen, Germany. [3] ARISTRA GmbH, Berlin, Germany. [4] The Institute for Experimental Molecular Imaging, RWTH Aachen University, Aachen, Germany. [5] Fraunhofer Institute for Digital Medicine MEVIS, Bremen, Germany. [6] Comprehensive Diagnostic Center Aachen (CDCA), University Hospital RWTH Aachen, Aachen, Germany. [7] These authors contributed equally: Volkmar Schulz, Daniel Truhn. ✉email: tianyu.han@pmi.rwth-aachen.de; schulz@pmi.rwth-aachen.de; dtruhn@ukaachen.de

Computer vision (CV) in medical imaging has been a focus of radiological research in recent years. It is likely that CV methods will soon be used as adjunct tools by radiologists: Computer-aided diagnosis can help to speed up the diagnostic process by guiding radiologists to findings worth looking at and maximize diagnostic accuracy by reducing subjectivity[1–5]. Prominent examples are deep convolutional neural networks (CNN), which had their breakthrough when more conventional computer vision algorithms were far surpassed by residual neural networks in 2015[6]. Similar developments have taken place in medicine, where CNNs performed comparable to experts in lung cancer diagnosis[7–9], retinal disease detection[10–12], and skin lesion classification[13–15]. However, certain problems in CV still exist: deep learning models trained in a standard fashion are vulnerable when facing attacks from adversaries. An attacker might introduce a subtle change into the image - such as changing a single pixel[16] - and manipulate the output of the model towards a desired direction.

To date, adversarial attacks have been of interest primarily to computer science researchers. With the landscape of competing interests within healthcare and with the continuing integration of machine learning in clinical practice, it is likely, that adversaries will arise that exploit such vulnerabilities. To illustrate possible scenarios: insurances could employ deep learning for approval of insurance claims, thus incentivizing adversaries which aim to commit insurance fraud. Another scenario might be a company seeking FDA approval for its newly developed drug in which the efficacy of said drug is measured with a radiological response (e.g., the shrinkage of tumor volume). The company might be tempted to manipulate the follow-up radiological images in an adversarial manner in order to promote its drug's approval[17]. For the clinician, it is thus important to understand, why a model arrives at a certain conclusion and if that reasoning aligns with human reasoning: A reliable and robust mechanism to explain the model's reasoning could support the acceptance of deep learning models in clinical routine[18]. Great effort has been devoted to techniques such as feature and attribution visualization to solve the concern described above. Techniques such as class activation maps, i.e., CAM[19] and GradCAM[20] visualize where the network focusses its attention to. Gradient based methods study, which pixels of the input image are responsible for the neural network firing in a particular way[21]. These methods can be applied irrespective of the way in which the neural network is trained.

Adversarial training offers an efficient way to both counteract adversarial influence and clarify the connection between input and output[22]. Nevertheless, in CV, researchers found that it is generally hard to obtain a both accurate and robust model though adversarial training[23]. In this paper, we address the mentioned degradation of accuracy via investigating loss landscapes of robust and non-robust models. More importantly, we find that training the model in this fashion allows for the model's reasoning to be more closely aligned with clinical expectations than when the model is trained in a conventional fashion. We test these hypotheses in dedicated experiments on X-rays, computer tomography images and magnetic resonance images and involve six radiologists who rate the clinical validity of our results.

## Results

**Adversarial training smoothes the loss surface.** Supervised models are vulnerable to adversarial attacks in which an adversary subtly changes the input to the model and thereby manipulates the prediction of that model. Instead of optimizing the parameters $\theta$, e.g., the weights of the neurons, of a model towards the minimum of the loss function $L$,

$$\min_{\theta} \; \mathop{\mathbb{E}}_{(x,y)\sim D} \left[ L(x,y;\theta) \right], \tag{1}$$

one is able to generate adversarial examples $(x + \delta)$ by solving the optimization problem

$$\mathop{\mathbb{E}}_{(x,y)\sim D} \left[ \max_{\delta \in \Delta} \; L(x+\delta, y; \theta) \right], \tag{2}$$

where $(x, y)$ is an input-label pair in the dataset D, $\delta$ is the applied adversarial perturbation, and $\Delta$ is an allowable set of perturbations. In practice, adversarial examples will always be designed to be as unconscious as possible to the human eye. One commonly define the allowed perturbations set $\Delta$ to be a hypersphere ball around any data $x$ with a constrained norm (e.g., $l_\infty \le \epsilon$)[24]. To select the best model in the task of pathology detection, we quantified the model's performance via the area under the receiver operating characteristic curve (ROC-AUC) and precision-recall curve. In accordance with previous research, we found that adversarial perturbations can easily influence conventionally trained models when applied to disease detection in thoracic X-rays, see Fig. 1.d: the standard classifier was significantly biased even by a small amount of adversarial perturbation ($\epsilon \le 0.01$). In a second experiment, we made the models robust to adversarial attacks by employing an approach proposed by Madry et al.[22]. In this approach, we minimized the expected adversarial loss via performing gradient descend on adversarial samples—effectively presenting the model with adversarial examples during training:

$$\min_{\theta} \; \mathop{\mathbb{E}}_{(x,y)\sim D} \left[ \max_{||\delta|| \le \epsilon} \; L(x+\delta, y; \theta) \right]. \tag{3}$$

Figure 1 visualizes our result, that an adversarially trained (robust) classifier (Fig. 1e) was less sensitive to adversarial perturbations than its counterpart that was trained in a standard fashion (Fig. 1d).

Nevertheless, other groups have previously shown, that robust models appear to be less accurate than standard models[23,25,26]. We tested these findings in the context of medical datasets by performing adversarial training on the Luna16[27], kneeMRI[28], and CheXpert[29] datasets (shown in Fig. 2 and Supplementary Fig. 1).

We found that robust models were indeed less accurate when trained on limited datasets, see blue and green curves in Fig. 2a, b. However, we found that the performance gap between standard and robust models was less pronounced when sufficient amounts of data were available, see Fig. 2c.

To understand the performance loss, let us consider an $\epsilon$ bounded perturbation. From Eq. (3), with weak perturbations $\delta$, one can expand the inner max function:

$$\max_{||\delta|| \le \epsilon} \; L(x+\delta, y) \approx L(x,y) + \max_{||\delta|| \le \epsilon} \left[ \delta \nabla_x L(x,y) + \frac{1}{2}\delta^\top H(L)\delta + \mathcal{O}(\delta^3) \right]$$
$$\approx L(x,y) + \delta^* \nabla_x L(x,y) + \frac{1}{2}\delta^{*\top} H(L)\delta^*, \tag{4}$$

where $\delta^*(x) = \mathrm{argmax}_{||\delta|| \le \epsilon} L(x+\delta, y)$. According to Eq. (4), the difference between a standard model and a model trained in an adversarial sense are the two additional terms containing $\delta$ (up to the third order in $\delta$). The tension between accuracy and robustness in adversarially trained models closely relates both terms. This can be understood as follows: to achieve higher robustness, i.e., $f(x + \delta) \approx f(x)$, adversarial training regularizes the model through minimizing its Jacobian and Hessian matrices (Eq. (4)). Such a regularization makes the model more invariant to all directions of perturbations: the loss surface is smoothed with

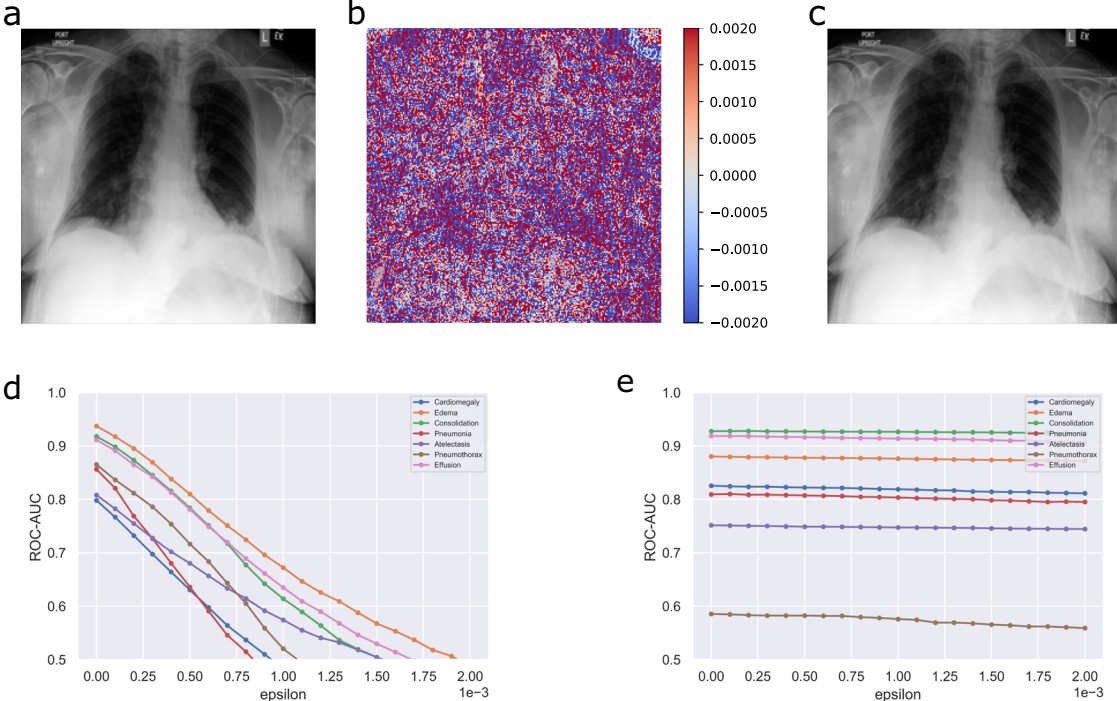

**Fig. 1 Adversarially trained models are robust against adversarial attacks.** Adversarial perturbations with increasing strength ($\epsilon$) were generated via an projected gradient descend (PGD) attack. To demonstrate the impact of adversarial attacks on state-of-the-art classifiers, we trained a ResNet-50 models with a large chest X-ray dataset (CheXpert) containing nearly 200,000 X-rays. **a** Original unmanipulated chest radiograph. **b** Adversarial noise with $\epsilon = 0.002$. **c** Manipulated chest radiograph (original radiograph + noise), i.e. adversarial example. **d** The standard model was easily misled by small adversarial perturbations (**b**) that are not perceptible to the human eye (**c**) and accuracy in classifying the disease dropped drastically when allowing more pronounced perturbations. **e** Only a limited amount of performance degradation was observed when applying adversarial attacks on the model trained adversarially ($\epsilon$ during training was set to 0.005).

respect to its inputs, see Supplementary Fig. 2. In Supplementary Table 6, the loss Lipschitz of the standard model is larger than that of the robust model indicating a higher adversarial vulnerability. Importantly, the model's sensitivity to non-robust but useful features[30] is limited due to such over-smoothing effect and therefore leads to accuracy degradation. In other words, training with adversarial images and only a single batch norm leads to over-smoothed loss surfaces that might miss important details that help in differentiating separate classes. This is different if we employ dual batch norms as demonstrated in the following section.

**Revisiting adversarially augmented training.** To balance the accuracy decrease in robust models, we treat adversarial examples as a form of augmentation and train our models with auxiliary batch norms[31–33]. Within this setting, we can formulate the training objective as:

$$\min_{\theta', \gamma, \gamma'} \mathbb{E}_{(x,y) \sim D} \left[ L(\theta', \gamma; x, y) + \max_{||\delta|| \leq \epsilon} L(\theta', \gamma'; x + \delta, y) \right], (\theta', \gamma, \gamma') \in \theta,$$

(5)

where $\gamma$ and $\gamma'$ represent parameters in $\text{BN}_{\text{std}}$ and $\text{BN}_{\text{adv}}$, whereas, $\theta'$ represents remaining parameters in the model. Under weak perturbations, we can expand and approximate the above objective as:

$$\min_{\theta} \mathbb{E}_{(x,y) \sim D} \left[ L(\theta, x, y) + \delta^* \nabla_x L(\theta', \gamma'; x, y) + \frac{1}{2} \delta^{*\top} H(L(\theta', \gamma')) \delta^* \right].$$

(6)

Here, $\theta$ in the first term of Eq. (6) contains all parameters of the network. As shown in Eq. (6), regularization terms only affect $\gamma'$

and $\theta'$. In general, batch norm layers are essential parts determining the model's performance and robustness. Making use of dual batch norms allows us to separately study the role of batch norm layers in adversarial training. In Supplementary Fig. 3, the loss surface changes dramatically when we switch from $\text{BN}_{\text{adv}}$ ($\gamma'$) to $\text{BN}_{\text{std}}$ ($\gamma$) while keeping the same set of parameters for the convolutional layers ($\theta'$). By keeping separate batch norms for the real and adversarial samples, we both avoid over-smoothing (as demonstrated in Supplementary Fig. 3a) and keep the robustness against adversarial samples (b in Supplementary Fig. 3). The loss surface of $\text{BN}_{\text{std}}$ is not over-smoothed by adversarial training and therefore preserves its accuracy. The above observation suggests a close link between batch norm layers and the loss landscape, i.e., smoothness and Lipschitzness, which is in agreement with the observations by Madry et al.[34].

In experiments with medical imaging, we observe, the employment of separate normalization layers removed the performance gap towards naively trained models: as demonstrated by the red AUC in Fig. 2, no performance difference towards naively trained models was found. A more detailed summary of performance metrics and confidence intervals can be found in Supplementary Tables 3–5. No significant differences in ROC-AUC, sensitivity, and specificity were found when comparing the naively trained non-robust model to the adversarially trained model with dual batch norm training. To verify that larger datasets and dual batch norms are necessary for accurate adversarial training, we randomly subsampled the CheXpert training set to 1% (1910 X-rays) and 10% (19,103 X-rays). Models were trained on the above CheXpert subsets and their performances on the test set are reported in Fig. 3. When incorporating more training data, we observe an ROC-AUC

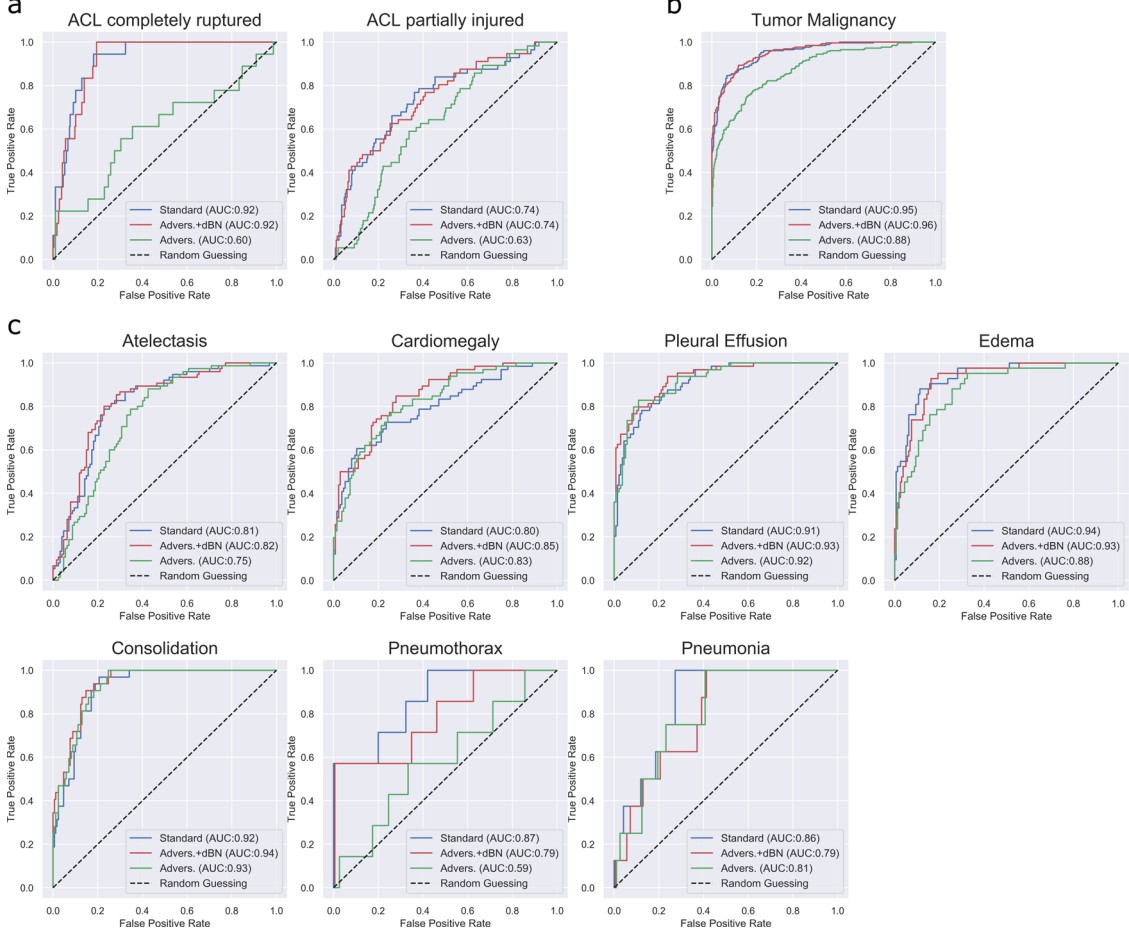

**Fig. 2 The usage of dual batch norm boosts the classification performance of neural networks.** Three models were compared: blue: neural network without adversarial training, green: neural network with adversarial training and red: neural network with adversarial training employing dual batch norms. The models' performances were tested on three distinct datasets: **a** Rijeka knee magnetic resonance imaging (MRI) dataset. **b** Luna16 dataset containing computed tomography (CT) slices of malignant tumors and (**c**) CheXPert thoracic X-ray dataset. We found that robustness and good performance appear to be incompatible when data is limited. In both experiments (**a** and **b**), the AUC of naively adversarially trained models (green) dropped significantly as compared to models trained in a standard fashion (blue). However that performance gap was reduced when the models were trained on a large dataset of 191,027 radiographs (**c**). Adversarially trained models performed best when employing dual batch norm (red), no significant difference in performance to the naively trained models were found. As reflected by the red curves, the performance of robust models was boosted across different datasets when dual batch norm training was employed (**a**–**c**).

increase in most pathology classifications in the setting of the adversarially trained dual batch norms model in accordance with our hypotheses. More particular, in the case of cardiomegaly, edema, and pneumothorax, our results show that the AUC of the conventionally trained (i.e., one batch norm) adversarial model remains comparatively low (even at 100% of the training set, performance is similar to the situation with 10% of the training set) while the AUC of the adversarial model with dual batch norms continues to increase when adding data. The adversarial model with dual batch norms generalizes the best in the task of cardiomegaly and effusion classification. However, in pneumonia classification, we also observe that the performance of adversarial models are both less accurate and less stable when comparing with their standard counterpart. One possible explanation is the limited number of pneumonia cases in the whole CheXpert dataset: only 2.4% of CheXpert are pneumonia positive[29], which might restrict the classification performance of adversarial training.

In practice, generalization across different domains of CV algorithms can be challenging. To explore whether this was the case with our robust training, we used the external ChestX-ray8 dataset as an additional test dataset. A comparison of the distribution of ground-truth labels of both datasets is listed in Supplementary Table 1. Figure 4 shows a comparison of the standard and robust models solely trained on the CheXpert dataset. Models were validated on the external ChestX-ray8 dataset that contained 22,433 radiographs and had never before been presented to the models[35]. As before, the robust model employing dual batch norms outperformed the conventional adversarially trained model and performed comparably to the standard model.

However, we also observe that a robust model even with dual batch norms still generalizes slightly worse than its standard counterpart when classifying pathologies such as cardiomegaly, edema, and atelectasis. By comparing with Fig. 3, such a behavior is different from the generalization on independent and identically distributed (IID) test sets of the CheXpert data. One contributing factor is the distribution shift between both datasets: although both datasets share similar pathological labels and regions of interests (ROI) on chests, a considerable shift in pixel intensity distribution still exists[36]. Another important factor is the sample complexity of robust learning, which is generally larger

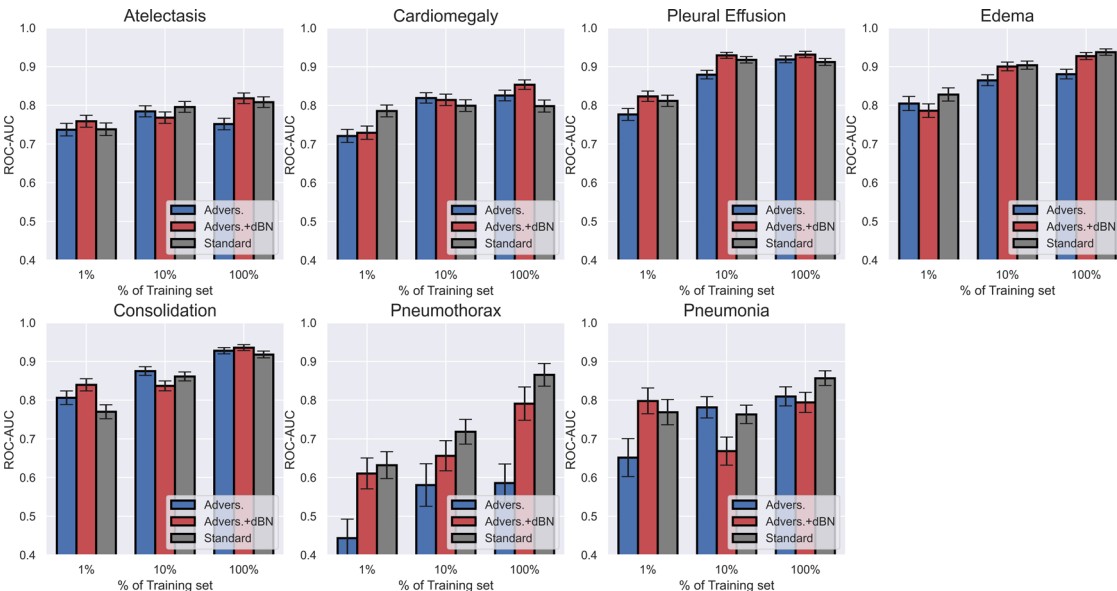

**Fig. 3 More training data and dual batch norms are essential to accurate adversarial training.** The classification performance of a standard, an adversarial (blue column), and an adversarially augmented model (red column)) with respect to different amounts of training data. In accordance with our hypotheses, the performance of adversarially trained models were boosted both by employing the dual batch norm and by enlarging the training set. In the case of pneumonia classification, the performance of adversarially trained models was limited and less stable due to an insufficient amount of pneumonia positive cases in the dataset. Data are presented as mean values $+/-$ SD (standard deviation). Note, $n = 10{,}000$ redraws are calculated in the bootstrapping analysis to get the mean and SD.

than the standard one[37] and therefore always requires more data to generalize (also reflected in Fig. 3).

**Adversarial training selects features that are close to what experts consider meaningful.** As previously reported by Tsipras et al.[23] loss gradients of robust models were found to be both sparse and well aligned with human expertize. In Fig. 5, it was found that the saliency maps of the adversarially trained neural network with dual batch norm (SDBN) agreed significantly better with human expertize than both the adversarially trained models with a single batch norm (SSBN) and those of the standard model (SSM): Six radiologists were given the task to rate the meaningfulness of the saliency maps in guiding the radiologist to the correct pathology on a scale from 0 (no correlation between pathology/ies and hot spot(s) on saliency map) to 5 (clear and unambiguous correlation between pathology/ies and hot spot(s) on saliency map), see Table 1 and Supplementary Table 2. More precisely, a robust model, as shown in red in Fig. 6a, was able to detect common thoracic diseases such as cardiomegaly, atelectasis, and pneumonia based on the organ shape and the lung opacity. While the SDBN pointed more clearly to the areas of interest that were decisive for diagnosis, the SSN had less focus on these areas and the SSM was almost completely uncorrelated to these areas and useless in guiding the radiologist to the correct conclusions. Similarly, for knee magnetic resonance images (Fig. 6b), the SDBN showed a more direct correlation with the pathology than the SSBN, which more often pointed to accompanying, but more unspecific phenomena such as joint effusion. Again, the SSM was almost completely uncorrelated to the imaging pattern of the disease. Finally, for the intrapulmonary malignancies in CT slices shown in Fig. 6c, the finding that SDBN, SSBN, and SSM were useful in descending order was again confirmed with the borders of the malignancy being emphasized more pronounced in the SDBN.

Interpretability of gradients is closely related to adversarial training itself and not attributable to a greater number of training images via augmentation. To demonstrate this, additional models were trained by augmenting the input training images with random pixel noise. Supplementary Figure 4, visualizes the loss gradient of models trained with medium and strong Gaussian noise augmentation, i.e., $\sigma = 0.01$ (1st row) and $\sigma = 0.1$ (2nd row), and another model with adversarial training (last row). Based on the rating standard in Table 2 of the manuscript, a radiologist assessed the diagnostic relevance of 100 randomly selected chest X-rays and their gradient saliencies. The evaluation scores were $1.14 \pm 1.23$, $1.35 \pm 1.37$, and $2.14 \pm 1.43$ for random noise augmentation ($\sigma = 0.01$ and $\sigma = 0.1$) and adversarial augmentation. Thus we confirmed that the gradients of the loss with respect to the input pixels were semantically meaningful and sparse in the adversarial model, whereas the saliency maps from pixel-noise augmented models were noisy and only loosely connected to occurring pathologies. The sparsity of the gradients can be in parts explained if we consider the case of generating $\delta^*$ via $l_\infty$ bounded fast gradient sign attacks:

$$\delta^* \nabla_x L(x, y) = \epsilon \operatorname{sign}(\nabla_x L(x, y)) \nabla_x L(x, y)$$
$$= \epsilon || \nabla_x L(x, y) ||_1. \tag{7}$$

Sparser Jacobians in Fig. 6 are obtained via minimizing the loss term above. Several studies have also demonstrated that the high interpretability of gradients is due to adversarial training, because adversarial training leads to confinement of gradients to an image manifold[30,38].

**Batch normalization influences learned representations.** In the setting of adversarial training with dual batch norms, original and perturbed batches are decoupled via passing them through separate batch norm layers. To understand how the use of these two batch norms influenced the features learned by the robust models, we quantified the similarity between the layers within the respective deep neural network by using the linear centered kernel alignment (Linear CKA) method[39]. In Fig. 7a, we visualized the typical representation learned by a model in a standard training setting, with only one batch norm. We found that a

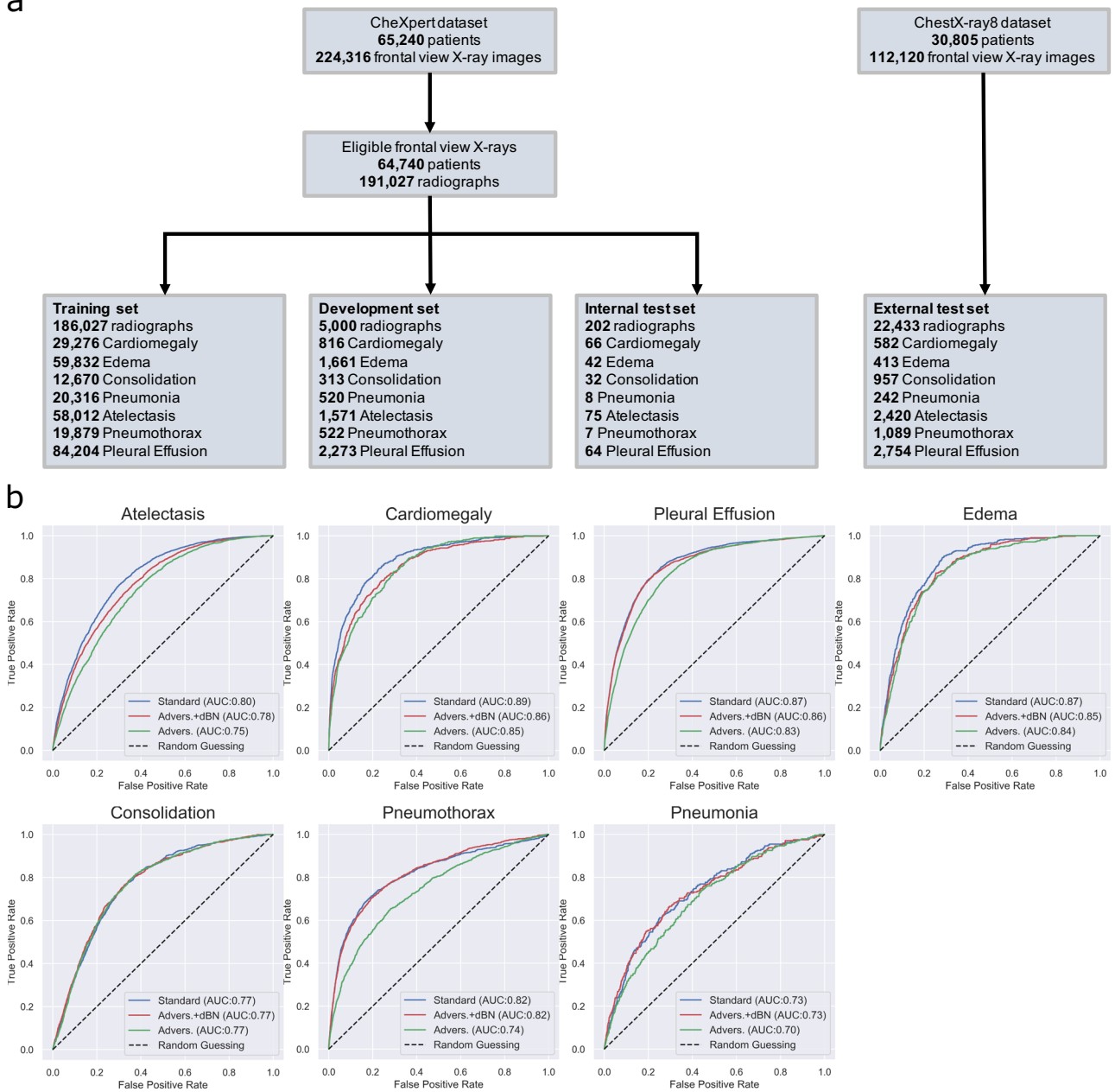

**Fig. 4 Validation of models on an external dataset (ChestX-ray8). a** Schematic of the data selection process. **b** AUC of the standard model (blue) and the adversarially trained models with (red) and without (green) dual batch norms on an independent test set of 22,433 radiographs from the ChestX-ray8 dataset. Dual batch norm training resulted in better AUC, closely matching the performance of the standard model.

certain degree of correlation between succeeding layers was present. However, long-range correlations—i.e., correlations between layers that were far apart—tended to be relatively weak, indicating, that the information that was passed on in the network gets continuously processed. The situation was different however for the same network architecture when adversarial training with only one batch norm was used, see Fig. 7b: Long ranging correlations resulted in a block like structure and the first 35 layers (about 65% of the network) seemed to carry approximately the same information. Such a high similarity of learned representations may be part of the reason for the performance degradation of robust models trained via vanilla adversarial training[23,39]: it seems that the networks may not be able to encompass the full complexity of the dataset after adversarial training. The network seems to effectively reduce to a simpler—less deep network since

neighboring layers contained similar activations. Using a dual set of batch norms for the original image samples and the adversarial image samples seemed to preserve the complexity of the network when fed with the original samples (Fig. 7c), while at the same time providing the same transition as in Fig. 7b for the adversarial samples as indicated by the similarity between the linear CKA maps of Fig. 7b, d.

## Discussion

The purpose of this study was to investigate the applicability and potential advantages of adversarially robust models in the field of medical imaging. A limitation of deploying such models in clinics is a potential performance degradation as compared to conventionally trained models that has been found by other research groups[23]. In our experiments shown in Fig. 2, however, we found

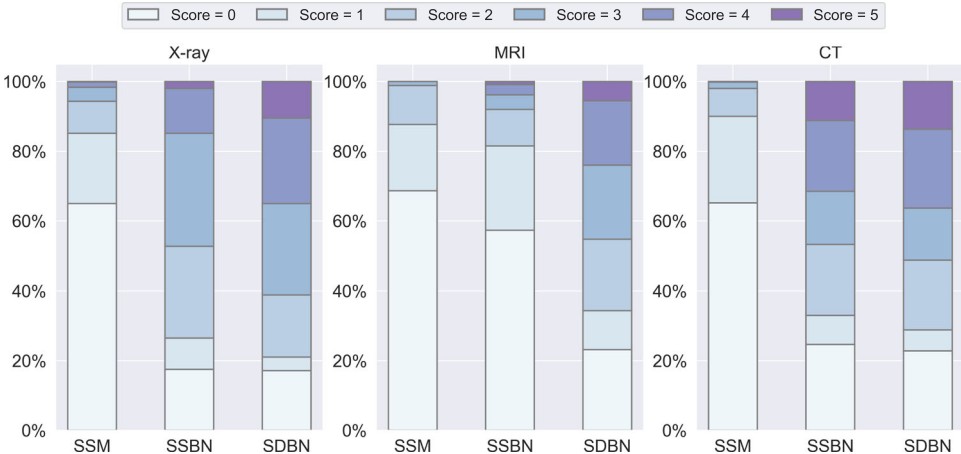

**Fig. 5 Adversarially trained neural network with dual batch norm yields clinically interpretable saliency maps.** Figure shows the assessment of diagnostic relevance in percentage for SSM, SSBN, and SDBN models as evaluated independently by six radiologists. Each color bar reveals percentage of gradient saliencies with same rating score.

| Table 1 Mean ratings of radiologists in guiding the radiologist to the correct pathology. | | | |
|---|---|---|---|
| **Dataset** | **Score for SSM** | **Score for SSBN** | **Score for SDBN** | **Friedman test** |
| X-ray | 0.57 ± 0.94 | 2.20 ± 1.33 | 2.69 ± 1.56 | $p < 0.001$ |
| MRI | 0.49 ± 0.74 | 0.74 ± 1.09 | 2.17 ± 1.57 | $p < 0.001$ |
| CT | 0.47 ± 0.73 | 2.32 ± 1.72 | 2.50 ± 1.74 | $p < 0.001$ |

The scale ranges from 0 (useless) to 5 (saliency map points clearly and unambiguously to the correct pathology). In total 100 images were rated by 6 radiologists for each dataset. One-sided Friedman tests were used to the $p$-value. Exact $p$-values are $1.0 \times 10^{-160}$, $1.7 \times 10^{-118}$, and $7.0 \times 10^{-150}$ for X-ray, MRI, and CT images.
*SSM* saliency maps of standard models, *SSBN* saliency maps of adversarially trained models with a single batch norm, *SDBN* saliency maps of adversarially trained models with dual batch norms.

that this effect appears almost negligible when training the models on large image data sets and when applying dual-batch norms, i.e., no significant difference in the AUC was found between the standard model and the adversarially trained model with dual batch norms. Furthermore, we have validated that robust models can generalize well on external datasets by employing 22,433 X-rays from the ChestX-ray8 dataset, that had not been part of the training process and originated from a different institution. Most likely, the reason that other groups had found significant differences between the performance of standard models and adversarially trained models is the use of a single batch norm in adversarial training: we consistently found in all our experiments, that to achieve the best results in adversarial training, it was necessary to employ separate batch norms for real and adversarial examples. In agreement with the results of Xie et al.[32] on RGB-images, we found that the use of dual batch norms can improve training and it allows our adversarially trained model to achieve state-of-the-art results on pathology detection.

Not only are adversarially trained models less vulnerable to adversarial attacks (see Fig. 1), but saliency maps generated by adversarially trained models provide significantly more information to the clinicians than those generated by standard models and may help to guide them to the right diagnosis. This can also boost the acceptance of deep learning models in clinical routine. Deep learning models are often regarded as a black box and not much trust is put into their opaque decision-making process by clinicians. By providing the clinician with a meaningful saliency map as generated by adversarially trained deep neural networks with dual batch norms, the decision of the neural network can be made more transparent resulting in a better acceptance by

experts. It should be noted however, that the question of whether these saliency maps indeed help to increase the endpoints of the diagnostic process, e.g. clinical reliability and accuracy, is yet an open research topic and an important subject for future work.

We further investigated a potential reason for the better performance of adversarially trained models with dual batch norms: while conventional adversarial training seems to reduce the complexity of the neural network as indicated by the increased long-range correlations of the linear CKA between layers of the network (see Fig. 7), the use of dual batch norms preserves complexity levels of the networks when feeding in real examples, while simultaneously accommodating for the increased robustness to adversarial examples.

This study is limited by the use of neural networks using two-dimensional inputs. Medical data in CT, MRI, and positron emission tomography (PET) is inherently volumetric (3D) or even volumetric plus time (4D) and it might be expected that models encompassing such higher dimensional inputs (instead of a series of two-dimensional slices) can improve upon their performance. More research is needed to train a model with high dimensional inputs as adversarial training commonly becomes more difficult in a high-dimensional feature space. If higher-dimensional models become more widespread and applicable, future studies should try to reproduce our results in such models. When applied to ImageNet classification tasks, the authors of[32] achieved a top-1 accuracy increase when performing adversarially augmented training. This finding is not reflected by our study, which matched, but not increased accuracy of the non-adversarially trained models. A potential reason for this might be the difference in dataset size and problem complexity: medical images are both rarer as a whole and greater in terms of pixel

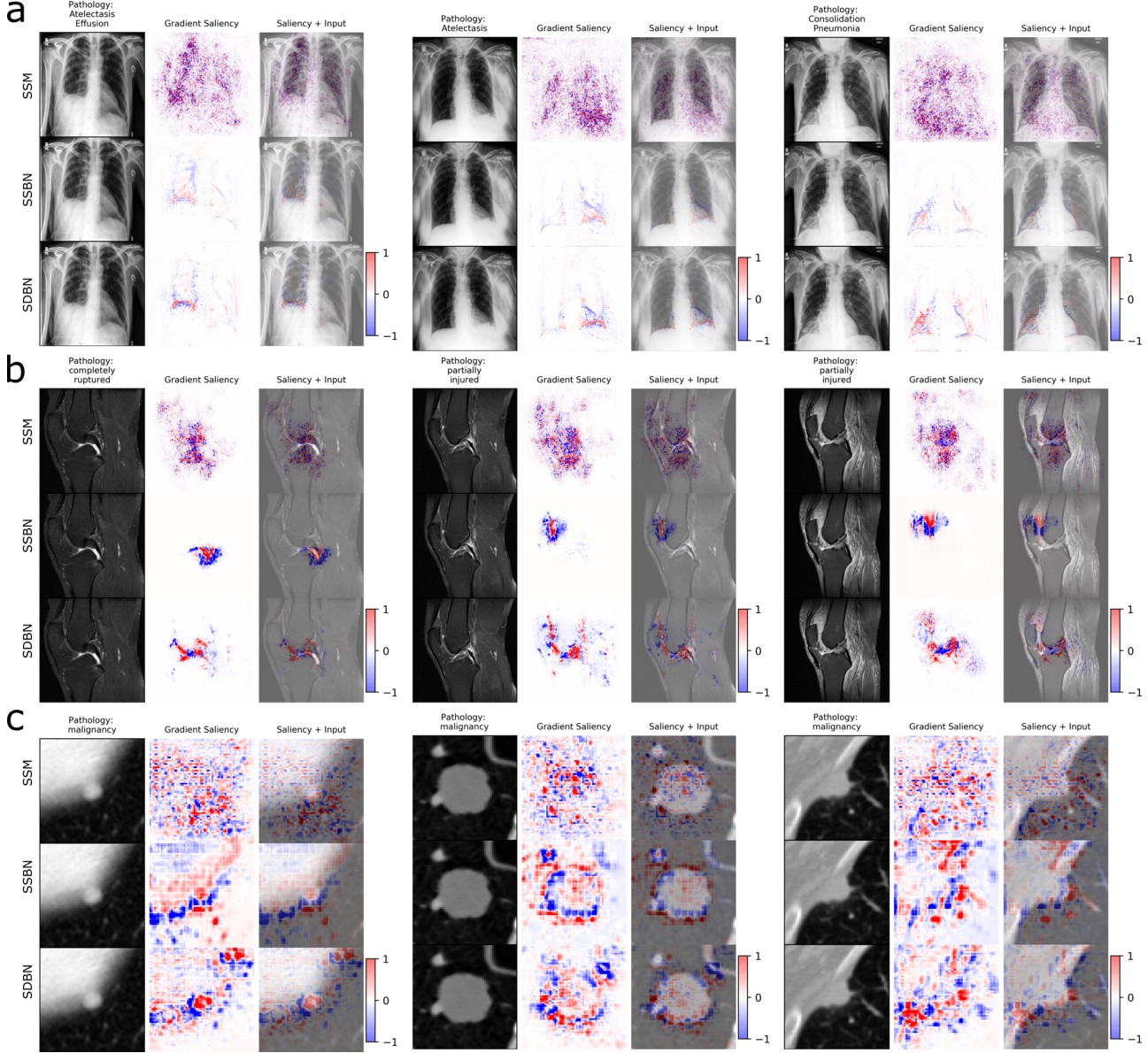

**Fig. 6 Saliency maps can help in guiding specialists to the correct diagnosis.** Loss gradients were plotted with respect to their input pixels for the X-ray (**a**), MRI (**b**), and CT (**c**) datasets. No extra preprocessing steps were applied to the loss gradients. Here, blue and red colored pixels denote negative and positive gradients individually. For comparison, saliency maps of standard and robust models are presented next to the radiological input. In all three datasets, the SDBN pointed more accurately and more distinctively to the pathology than the SSBN, while the SSM was almost completely uncorrelated and almost noise-like. It can also be observed that the saliency maps created based on neural networks that have been trained on a very large dataset of hundreds of thousands of images (**a**) were more precise in pointing to the pathology, than those trained on datasets containing fewer examples (**b** and **c**).

numbers individually, making the problem considerably harder. However, the results of[31–33] and our results in the experiments with reduced data indicate, that accuracy of adversarially trained models with separate batch norms might surpass the standard networks if sufficient amount of data is available.

In conclusion, we demonstrated, that adversarially trained models with a dual batch norm are not only equivalent to standard models in terms of diagnostic performance but offer additional advantages in conveying their reasoning through the use of clinically useful saliency maps and being more robust to adversarial attacks. We encourage fellow research groups to employ adversarially trained neural networks in their applications and hope that this will not only lead to more robust and better results

in terms of diagnostic performance, but also increased acceptance of such algorithms in clinical practice.

## Methods

**Study datasets**. A total number of four medical imaging datasets are used in this study: the CheXpert dataset, which has been released by Irvin et al. in January 2019 and contains 224,316 chest radiographs of 65,240 patients[29]. Only 191,027 frontal radiographs are downloadable for model training. To clean up CheXpert labels, we assigned pathology labels not mentioned to 0.0. According to the labeling performance comparison[29], the uncertainty labels (U) were assigned to 1.0, except for the consolidation class (to 0.0). For testing, we compare the performance of the trained models on the official validation set of 202 scans on which the concurrence of diagnosis from three radiologists serves as ground truth[29]. Another X-ray dataset used in this study is the ChestX-ray8 dataset released by the National Institutes of Health (NIH) in 2017, containing 112,120 frontal radiographs of 30,805 unique

patients[35]. We randomly select 20% out of 112,120 radiographs, i.e., 6187 patients and 22.433 radiographs, to form an external test set. The seven overlapping labels between the CheXpert and ChestX-ray8 datasets are listed in Supplemtary Table 1.

**Table 2 Rating standard used for evaluating the diagnostic value of generated saliency maps.**

| Score | Diagnostic Rating |
|---|---|
| 0 | No correlation between pathology/ies and hot spot(s) on saliency map. |
| 1 | Highly doubtful low-degree correlation between pathology/ies and hot spot(s) on saliency map. |
| 2 | Doubtful moderate-degree correlation between pathology/ies and hot spot(s) on saliency map. |
| 3 | Definite partial correlation between pathology/ies and hot spot(s) on saliency map. |
| 4 | Definite substantial correlation between pathology/ies and hot spot(s) on saliency map. |
| 5 | Clear and unambiguous correlation between pathology/ies and hot spot(s) on saliency map. |

The MRI dataset used in this study is the kneeMRI dataset, which has been released by Štajduhar et al.[28] in 2017 and contains 917 sagittal proton-density weighted knee scans from Clinical Hospital Centre Rijeka, Croatia. Three degrees of anterior cruciate ligament (ACL) injuries were recorded by radiologists in the Rijeka dataset: namely non-injured (692 scans), partially injured (172 scans), and completely ruptured (55 scans). Bounding box annotations of the region of interest (ROI) slices responsible for ACL tear diagnosis were also provided along with labels. According to ROI labels, we extracted a total of 3081 diagnostic relevant MRI slices and randomly split them into 80% training, 10% development, and 10% testing. Lastly, we investigated the applicability of adversarial training on CT data with the Luna16 dataset consisting of 888 CT scans with lung cancer ROI annotations[27]. In total, a number of 6,691 lung cancer patches were extracted and randomly split into 80% training, 10% development, and 10% testing.

**PGD attack and adversarial training**. Following Eq. (1), we let the model parameter be denoted as $\theta$, model loss as $L$ and training input&label as (x, y). The projected gradient descent (PGD) method repeatedly adjusts the model's inputs x in the direction of maximizing the loss function, i.e., $sign(\nabla_x L(x, y; \theta))$. To safeguard models against adversarial threats, we trained our models against a PGD adversary via both vanilla and dual batch norm adversarial training. The details of the adversarial training procedure with separate batch norms are depicted in algorithm 1. We plot the binary cross-entropy loss projected along two directions, i.e., the adversarial ($\epsilon_{\nabla_x L}$) and a random ($\epsilon_{Rad}$) direction, for samples in CheXpert test set. The grid size in Supplementary Figs. 2 and 3 is $50 \times 50$.

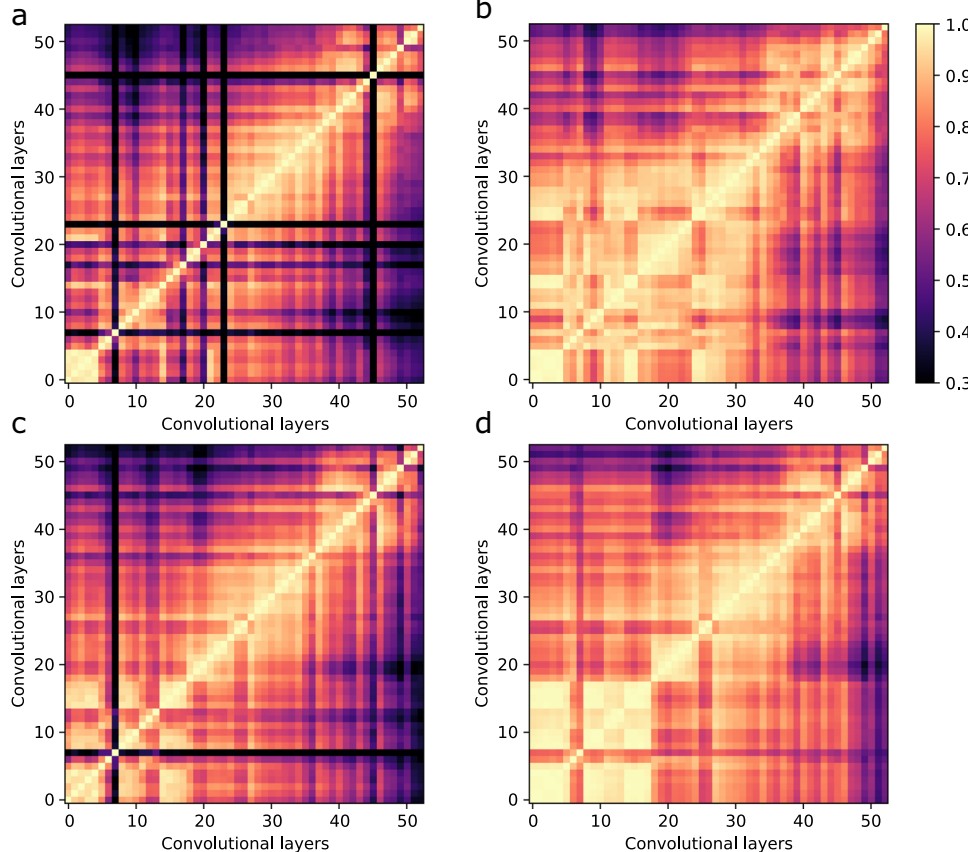

**Fig. 7 Linear centered kernel alignment (CKA) reveals representations are influenced by batch norms.** To explore the learned hidden representations, the linear CKA between convolutional layers of the models was computed on the CheXpert test set: a model trained with a single batch norm in a conventional setting with real examples (**a**), a model trained with a single batch norm with real and adversarial examples (**b**), and a model trained with a dual batch norm with real and adversarial examples when the respective CKA was evaluated separately with the batch norm used for real (**c**) and adversarial (**d**) examples. It should be noted that the observed grid pattern in **a** was due to the residual connections in the ResNet architecture[39]. When employing adversarial training with a single batch norm, layers of the network seem to get more similar to each other, as visualized by the block-like structure arising from the high degree of similar neural activations in (**b**). This indicates, that the neural network loses complexity due to adversarial training which might contribute to a loss in performance. When employing a dual batch norm for original and adversarial examples respectively, the complexity of the network seems to be preserved (note the similarity between **a** and **c**), when presented real examples using the first batch norm, while simultaneously robustness to adversarial examples arises due to the same changes when employing the second batch norm (**d**) that the network from (**b**) underwent (note the similarity between **b** and **d**).

## Algorithm 1

**Adversarial Training with Separate BN_std and BN_adv.** We use default values of $\epsilon = 0.005$, $\alpha = 0.0025$, $k = 10$, and $b = 64$

1: **Require:** Dataset D; A model h with its current parameter $\theta$ and loss function L; batch norm layers for standard inputs $BN_{std}$; batch norm layers for adversarial inputs $BN_{adv}$; the batch size b; learning rate $\eta$.

2: **Require:** $l_\infty$ boundary $\epsilon$; step-size $\alpha$; number of attack iterations k.

3: Sample a batch of inputs $\{x^{(j)}\}_{j=1}^b \sim D$ and class labels $\{y^{(j)}\}_{j=1}^b \sim D$.

4: **for** $j = 1, ..., b$ **do**

5:     Sample input $x \sim \{x\}^b$ and label $y \sim \{x\}^b$

6:     $x_0^* = x$

7:     **for** $t = 1, ..., k$ **do**

8:         Input $x_{t-1}^*$ to model

9:         $x_t^* \leftarrow x_{t-1}^* + \alpha \, \text{sign}\,(\nabla_x(x_{t-1}^*, y; \theta))$

10:        $x_t^* \leftarrow \text{clip}_{x,\epsilon}(x_t^*)$

11:    **end for**

12: **end for**

13: $\{x^*\}^b = \{x_k^*\}^b$

14: $L_{std} = L(\{x\}^b, \{y\}^b; \theta, BN_{std})$; $L_{adv} = L(\{x^*\}^b, \{y\}^b; \theta, BN_{adv})$

15: $\theta \leftarrow \theta + \eta \nabla_\theta(L_{std}^j + L_{adv}^j)$

**Model architecture and training**. We used ResNet-50 architecture for our experiments in this study. For all classification tasks, an Adam optimizer with default $\beta_1 = 0.9$, $\beta_2 = 0.99$, and $\epsilon = 1e-8$[40] was used to optimize the loss. In a total number of 300 training epochs, We decayed the initial learning rate 0.01 by a factor of 10 once the number of epochs reached 100 epochs. All classifier models utilized development-based early stopping with sigmoid binary cross-entropy loss as the criterion.

Medical images from CheXpert, ChestX-ray8, and kneeMRI were scaled to a fixed resolution of $256 \times 256$ pixels whereas tumor patches extracted from Luna16 ROI slices were scaled to $64 \times 64$ pixels. During training, random color transformations such as adjusting contrast, brightness, saturation, and hue factor were applied to each training image. In addition, we also performed spatial affine and random cropping augmentations before normalizing each input to the range of 0 to 1.

All computations were performed on a GPU cluster equipped with two Intel Xeon(R) Silver 4208 processor (Intel, Santa Clara, Calif) and three Nvidia Titan RTX 24 GB GPUs (Nvidia, Santa Clara, Calif). When not otherwise specified, code implementations were in-house developments based on python 3.6.5 (https://www.python.org) and on the software modules Numpy 1.16.0, Scipy 1.21.0, and Pytorch 1.1.0.

**Model interpretation**. To reveal the connection between input features (pixels) and the model's final predictions, we back-propagate the loss gradients with respect to their input pixels. For all generated gradients, we simply clipped their values to the range of $\pm 3 \times$ standard deviation around their mean value and normalized them to $[-1, 1]$[23].

To investigate the interaction between learned representations in the deep neural networks and their batch norm layers, we quantified representation similarity via the linear centered kernel alignment (linear CKA). For a given input and a model, a linear CKA is defined as:

$$\text{CKA}(X, Y) = \frac{||Y^T X||_F^2}{(||X^T X||_F ||Y^T Y||_F)}, \quad (8)$$

where $X$ and $Y$ correspond to a centered Gram matrix of layer activations. In Fig. 7, we computed the linear CKA matrix across all 202 radiographs from the internal CheXpert test set.

**Feature evaluation by radiologists**. To evaluate the clinical utility of the generated saliency maps for the three models (standard model, adversarially trained model with a single batch norm, and adversarially trained model with the dual batch norm), we randomly chose 100 images from each of the three datasets used in this study (in total 300 images) and let six radiologists assess how useful the map was in guiding a radiologist to the correct diagnosis. We used a scale from zero, signifying no correlation between the pathology and the saliency map, to five, signifying a map that points clearly and unambiguously to the correct pathology—or pathologies if multiple pathologies were present in the image, see Table 2. All readers performed the task independently of each other.

**Statistical analysis**. For each of the experiments, we calculated the following parameters on the test set: area under the curve (AUC) for the receiver operator characteristic (ROC), sensitivity, and specificity. The cutoff value for deciding between the presence or non-presence of a pathology was determined by minimizing $(1 - \text{sensitivity})^2 + (1 - \text{specificity})^2$[41]. To assess errors due to sampling of the specific test set and estimate the confidence intervals we employed bootstrap analysis with 10,000 redraws. The difference in metrics, such as AUC, sensitivity, and specificity, was defined as a $\Delta$metric. For the total number of $N = 1000$ bootstrapping, models were built after randomly permuting predictions of two classifiers, and metric's differences $\Delta$metric$_i$ were computed from their respective

scores. We obtained the $p$ value of individual metrics by counting all $\Delta$metric$_i$ above the threshold $\Delta$metric. Statistical significance was defined as $P < 0.001$. To determine if differences were significant in the reader studies we employed the Friedman test to test for the presence of differences within the three groups. If the Friedman test was significant we tested for pairwise differences (i.e. SSM vs. SSBN, SSM vs. SDBN, and SSBN vs. SDBN) employing the Wilcoxon signed-rank test.

**Reporting summary**. Further information on research design is available in the Nature Research Reporting Summary linked to this article.

## Data availability

The two X-ray datasets used in this study are available in the NIH ChestX-ray8 database and Stanford CheXpert database under accession code https://nihcc.app.box.com/v/ChestXray-NIHCC and https://stanfordmlgroup.github.io/competitions/chexpert. The MRI and CT datasets are available in Rijeka knee MRI database and LUNA16 database under accession code http://www.riteh.uniri.hr/~istajduh/projects/kneeMRI/ and https://luna16.grand-challenge.org/Data/. In addition, we use the publicly available MNIST dataset http://yann.lecun.com/exdb/mnist/ for experiments in the Supplementary section. All data needed to evaluate the findings in the paper are presented in the paper and/or the supplementary material. Additional data related to this paper such as the detailed reader test data maybe requested from the authors.

## Code availability

Details of the implementation, as well as the full code producing the results of this paper, are made publicly available under https://github.com/peterhan91/Medical-Robust-Training and via Zenodo (https://doi.org/10.5281/zenodo.4926118).

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

## Author contributions

T.H. and D.T. devised the concept of the study, D.T., S.N., M.S., M.Z., F.P., and M.H. performed the reader tests. T.H. wrote the code and performed the accuracy studies. T.H. and D.T. did the statistical analysis. T.H., D.T., and V.S. wrote the first draft of the manuscript. All authors contributed to correcting the manuscript.

## Funding

## Competing interests

The authors declare no competing interests.

## Additional information

**Supplementary information** The online version available at https://doi.org/10.1038/s41467-021-24464-3.

