## [Peer Review File · Nature Communications]

Reviewers' Comments:

Reviewer #1:

Remarks to the Author:

This paper addresses the application of machine learning methods for medical image diagnosis. Several key and hot points are covered including adversarial robustness, generalisability, interpretability, with the methodology of adversarial training through dual batch normalisation. Three public medical image datasets are used for validation. Experimental results are rich and statistical analysis are well presented. However, there are several important issues not clearly investigated. The detailed questions are in the following.

- The technical novelty is limited, given applying the existing adversarial training methods and separate batch normalisation layers (the cvpr paper). There is also no mathematical proof or analysis on why such dual batch normalisation helps improve adversarial training. This makes the technical contribution of the paper is relatively weak.

- How the model is tested in practice? When a new testing sample comes in, only bn_std used? If yes, will this impede the model robustness?

- A major agreement made in this paper is that, sufficiently large training datasets help combat trade-off between robustness and model performance. However, the small datasets are trained on lung images and knee images, the large dataset is on chest x-ray images. The difference in datasets/tasks may also affect the experimental results, therefore, arriving in the claim with such non-controlled experimental setting is not convincing. Adding further experiments by progressively increasing data-scale on x-ray dataset will help validate this point.

- Generalisation on robust models section, the model is trained on chexpert and tested on chest8. Though the adversarial w/ dual bn generalize better than w/o dual bn, the performance of adversarial models is still worse than standard models. This experimental observation should be discussed.

- Following above question, the model test performance should also be reported on chexpert dataset, in order to reflect the generalisation gap on new domain.

- Arguing the visualisation saliency map as a kind of interpretation is somewhat over-claiming. It is more or less a kind of visualised attention map, which is not new actually. Though human-in-the-loop study involves radiologist to evaluate the meaningfulness of such maps, whether these maps indeed help diagnosis, in terms of accuracy or reliability, is unclear and is unlikely to happen in practice.

- Table S3,4,5 should also include the numerical results of robust model w/o batch norms, for clear comparisons.

- Real clinical usefulness and practice value of the current work is limited and not sufficiently validated in the paper. These limitations should be discussed.

Reviewer #2:

Remarks to the Author:

The paper evaluates how an adversarial training approach can improve usability of a deep residual convolutional neural network (CNN) architecture for medical image classification. The three key observations of the paper are:

(1) indeed adversarial attacks can be averted to some degree by this approach;

(2) dual batch normalisation helps to alleviate the accuracy decrease typically associated with

adversarial training;

(3) with adversarial training the usability of saliency maps marking relevant image areas for classification is improved.

Adversarial attacks exploit slightly „wrinkled“ CNN decision boundaries in the image space to identify directions where imperceptible manipulations of the image can lead to misclassifications. The adversarial training strategy employed in this paper anticipates these attacks during training, and adds correspondingly generated correctly labeled images to the training set. This leads to a more robust model, but typically decreases its classification accuracy slightly. Dual batch normalisation can counter this deterioration. The paper shows that adversarial training and dual batch normalisation work as expected on different medical image classification data sets.

The most interesting and novel observation is that compared to standard training, adversarial training yields saliency maps that are closer to what experts consider relevant image regions. From the results, I don't fully understand if that is indeed an effect of adversarial training, or rather of an increased number of augmented training examples. To clarify this issue one should compare the saliency maps resulting from standard augmentation with e.g., pixel noise with those resulting from adversarial training.

A few comments:

In the introduction, the authors mention that state of the art deep learning models behave like black-boxes. This is a bit confusing and should be clarified. There are several established approaches that help illuminating their inner workings: CAM (B. Zhou et al. 2016 Learning Deep Features for Discriminative Localization), GradCAM (Selvaraju et al. 2017 Class Activation Mapping GradCAM), guided backpropagation (Springenberg et al. 2014 Striving for Simplicity: The All Convolutional Net), or uncertainty (Gal et al. 2016. "Dropout as a bayesian approximation: Representing model uncertainty in deep learning.") are some examples. The authors should discuss their relationship to the present work, since they use and cite an established technique, to show how adversarial training improves saliency maps.

Please be more specific on the real danger of adversarial attacks in medical imaging. Are there scenarios, or prior results on when and how likely this can happen? This would add to the motivation of the paper.

The introduction suggests that adversarial training reduces the risk of networks clinging on to so-called „clever hans“ strategies, where trivial confounds in the imaging are exploited for classification. It is not clear from the results that this is the case. Please clarify, evaluate, or omit.

In the results section, the use of AUC-ROC is helpful but slightly limited for unbalanced data sets, such as pathology detection. Precision-recall analysis offers a more informative assessment of the classification results in these scenarios.

Dual batch norm strategy is good and has been recently described as an approach to counter accuracy degradation due to adversarial learning, e.g., in Wang et al. NIPS 2020, „Once-for-All Adversarial Training: In-Situ Tradeoff between Robustness and Accuracy for Free“

Xie et al. CVPR 2020 „Adversarial examples improve image recognition“

Xie et al. ICLR 2020. „Intriguing properties of adversarial training at scale“ (already cited by the authors, in its arxiv version)

These paper should be cited, and the relationship should be commented on.

The correlation between layers (can it be viewed as a collapse?) observed in models resulting from adversarial training, and the resolution of this correlation by dual batch training is a very interesting and insightful observation. Could you please discuss how this relates to prior work on

adversarial training and dual batch normalisation.

Evaluating the saliency maps by expert assessment is a good experiment.

Please evaluate if the improvement of the saliency maps is due to an increased number of augmented training examples, that would be also achieved if augmentation was done with pixel noise, or if it is truly linked to the adversarial strategy. This is not clear. Please also compare with other state of the art approaches that high-light relevant image areas such as e.g., GradCAM.

Response to the reviewers

Reviewer 1 Comments

Reviewer 1: Comment 1

This paper addresses the application of machine learning methods for medical image diagnosis. Several key and hot points are covered including adversarial robustness, generalisability, interpretability, with the methodology of adversarial training through dual batch normalisation. Three public medical image datasets are used for validation. Experimental results are rich and statistical analysis are well presented. However, there are several important issues not clearly investigated. The detailed questions are in the following.

Thank you very much for taking the time to review our manuscript and for your overall appreciation of its strengths. We hope to have sufficiently addressed all issues. Please see our detailed response to your comments below.

Reviewer 1: Comment 2

The technical novelty is limited, given applying the existing adversarial training methods and separate batch normalisation layers (the cvpr paper). There is also no mathematical proof or analysis on why such dual batch normalisation helps improves adversarial training. This makes the technical contribution of the paper is relatively weak.

Thank you very much for your guidance on how to improve the quality of our manuscript. Indeed, Xie et al. proposed to use separate batch normalisation layers in adversarial training to improve image recognition in their CVPR paper [1]. In our paper we address both the advancement in diagnostic performance and the improved explainability. To guide the attentive reader to the related work by Xie et al., we changed the following paragraph in our paper in the discussion section:

In agreement with the results of Xie et al. [1] on RGB-images, we found that the use of dual batch norms can improve training and it allows our adversarially trained model to achieve state-of-the-art results on pathology detection.

A detailed mathematical investigation on why training with separate batch norms helps to improve general adversarial training has not been given by any group so far to our knowledge. Intuition of why this is the case can stem from domain segregation issues [1, 2]. However, we address the above point through the lens of optimization and re-parameterization offered by batch norms. The following sections are added to the results section:

Adversarial training smoothes the loss surface

Let's first consider an ϵ bounded adversarial attack. An adversarially robust model is built via adversarial training:

$$\min_{\theta} \mathbb{E}_{(x,y) \sim D} \left[\max_{\|\delta\| \leq \epsilon} \mathcal{L}(x + \delta, y) \right] \quad (1)$$

With weak perturbations δ , one can expand the inner max function:

$$\begin{aligned} \max_{\|\delta\| \leq \epsilon} \mathcal{L}(x + \delta, y) &\approx \mathcal{L}(x, y) + \max_{\|\delta\| \leq \epsilon} \left[\delta \nabla_x \mathcal{L}(x, y) + \frac{1}{2} \delta^\top H(\mathcal{L}) \delta + \mathcal{O}(\delta^3) \right] \\ &\approx \mathcal{L}(x, y) + \delta^* \nabla_x \mathcal{L}(x, y) + \frac{1}{2} \delta^{*\top} H(\mathcal{L}) \delta^* \end{aligned} \quad (2)$$

where $\delta^*(x) = \operatorname{argmax}_{\|\delta\| \leq \epsilon} \mathcal{L}(x + \delta, y)$. According to 2, the difference between a standard model and a model trained in an adversarial sense is the first-order and second-order term of δ (up to the third order in δ). The tension between accuracy and robustness in adversarially trained models also closely relates both terms: To achieve higher robustness, i.e., $f(x + \delta) \approx f(x)$, adversarial training regularizes the model through minimizing its Jacobian and Hessian matrices (equation 2). Such a regularization makes the model more invariant to all directions of perturbations: the loss surface is smoothed with respect to its inputs, see Fig R1. In table R1, the loss Lipschitz of the standard model is much larger than the robust model indicating a higher adversarial vulnerability. Importantly, the model’s sensitivity to non-robust but useful features [3] is limited due to such over-smoothing effect and therefore leads to accuracy degradation. In other words, training with adversarial images and only a single batch norm leads to oversmoothed loss surfaces that might miss important details that help in differentiating separate classes. This is different if we employ dual batch norms as shown in the following section.

Fig. R1. Note that this figure corresponds to the figure S2 in the original manuscript. Adversarial training regularizes the model via minimizing its Jacobian and Hessian matrices. The loss surface of a model that is only trained on unaltered images (no adversarial augmentation, i.e. model trained in a standard manner) (A) and a model trained solely with adversarial images with $\epsilon=0.005$ (B) for the CheXpert dataset on four test images. We display the cross-entropy loss projected on one random ϵ_{Rad} and one gradient $\epsilon_{\nabla_x \mathcal{L}}$ direction in the input space. Due to regularization, the robust model B has a loss that is smooth both in the gradient direction and in the random direction, whereas the loss surface of the standard model A changes rapidly both in all directions.

Table R1: Lipschitzness of models on CheXpert test set

model name	Lipschitz constant of loss
Standard	0.706 ± 0.950
Advers.	0.360 ± 0.506

Revisiting adversarially augmented training

To balance the accuracy decrease in robust models, we treat adversarial examples as a form of augmentation and train our models with auxiliary batch norms. Under the above setting, we can formulate the training objective as:

$$\min_{\theta', \gamma, \gamma'} \mathbb{E}_{(x, y) \sim D} \left[\mathcal{L}(\theta', \gamma; x, y) + \max_{\|\delta\| \leq \epsilon} \mathcal{L}(\theta', \gamma'; x + \delta, y) \right], (\theta', \gamma, \gamma') \in \theta \quad (3)$$

where γ and γ' represent parameters in BN_{std} and BN_{adv} , whereas, θ' represents remaining parameters in the model. Under weak perturbations, we can expand and approximate the above objective as:

$$\min_{\theta} \mathbb{E}_{(x, y) \sim D} \left[\mathcal{L}(\theta, x, y) + \delta^* \nabla_x \mathcal{L}(\theta', \gamma'; x, y) + \frac{1}{2} \delta^{*\top} H(\mathcal{L}(\theta', \gamma')) \delta^* \right] \quad (4)$$

As shown in equation 4, regularization terms only affect γ' and θ' . In general, batch norm layers are essential parts determining the model’s performance and robustness. Making use of dual batch norms allows us to separately study the role of batch norm layers in adversarial training. In Fig. R2, the loss surface changes dramatically when we switch from BN_{adv} (γ') to BN_{std} (γ) while keeping the same set of parameters for the convolutional layers (θ'). By keeping separate batch norms for the real and adversarial samples, we both avoid oversmoothing (as demonstrated in Fig. R2 A) and keep the robustness against adversarial samples (B in Fig. R2). The loss surface of BN_{std} is not over-smoothed by adversarial training and therefore preserves its accuracy. The above observation suggests a close link between batch norm layers and the loss landscape, i.e., smoothness and Lipschitzness, which is in agreement with the observations by Madry, et al [4].

To convey our reasoning to the interested reader, we add the corresponding technical details in the Materials and Methods section:

Loss surface visualization

We plot the binary cross-entropy loss projected along two directions, i.e., the adversarial ($\epsilon_{\nabla_x \mathcal{L}}$) and a random (ϵ_{Rad}) direction, for samples in CheXpert test set. The grid size in Fig. R1 and R2 is 50×50 .

Fig. R2. *Note that this figure corresponds to the figure S3 in the original manuscript*
Loss landscape relates closely to batch norm layers. The loss surface of an adversarially augmented model ($\epsilon=0.005$) with BN_{std} (A) and BN_{adv} (B) for CheXpert on four test images. As a result of different reparameterization (γ or γ'), the loss surfaces are different in (A) and (B).

Reviewer 1: Comment 3

How the model is tested in practice? When a new testing sample comes in, only bn_{std} used? If yes, will this impede the model robustness?

We thank the reviewer for this question. Yes, this is correct. When a new testing sample comes in, only bn_{std} is used, see also Fig. 2 and 4 of the original manuscript.

To answer the question if this impedes the model's robustness, we investigated the robustness of the adversarially augmented models BN_{std} and BN_{adv} (training $\epsilon=0.005$) for CheXpert, Rijeka, and Luna datasets. The following section is added to the appendix of our manuscript:

We investigated the robustness of the adversarially augmented models BN_{std} and BN_{adv} (training $\epsilon=0.005$) for CheXpert, Rijeka, and Luna datasets. The trained parameters of BN_{std} (γ) are always kept secret while parameters of BN_{adv} (γ') and remaining parameters (θ') are available to the public. Here, we applied black and white box attacks to BN_{std} and BN_{adv} separately as PGD adversaries are exclusively generated via robust batch norms, e.g., γ' and θ' . Indeed we found, that the performance of BN_{std} remains acceptable when facing transferred PGD attacks with reasonable amplitudes (ϵ up to 0.01), see Fig. R3.

Fig. R3. Note that this figure corresponds to the figure S6 in the original manuscript

Robustness of adversarially augmented models. The classification performance of an adversarially augmented model ($\epsilon=0.005$) with BN_{adv} (1st row) and BN_{std} (2nd row) on CheXpert (A), Rijeka (B), and LUNA (C) test sets. The trained BN_{std} branch remains robust while facing transferred attacks from BN_{adv} .

Reviewer 1: Comment 4

A major agreement made in this paper is that, sufficiently large training datasets help combat trade-off between robustness and model performance. However, the small datasets are trained on lung images and knee images, the large dataset is on chest x-ray images. The difference in datasets/tasks may also affect the experimental results, therefore, arriving in the claim with such non-controlled experimental setting is not convincing. Adding further experiments by progressively increasing data-scale on x-ray dataset will help validate this point.

We thank the reviewer for this helpful comment. Additional experiments with sub-sampled training sets are performed and the corresponding results are included in the results section:

To verify that larger datasets and dual batch norms are necessary for accurate adversarial training, we randomly subsampled the CheXpert training set to 1% (1,910 X-rays) and 10% (19,103 X-rays). Models were trained on the above CheXpert subsets and their performances on the test set are reported in Fig. R4. When incorporating more training data, we observe an ROC-AUC increase in most pathology classifications in the setting of the adversarially trained dual batch norms model in accordance with our hypotheses. More particular, in the case of cardiomegaly, edema, and pneumothorax, our results show that the AUC of the conventionally trained (i.e. one batch norm) adversarial model remains comparatively low (even at 100% of the training set, performance is similar to the situation with 10% of the training set) while the AUC of the adversarial model with dual batch norms continues to increase when adding data. The adversarial model with dual batch norms generalizes the best in the task of cardiomegaly and effusion classification. However, in pneumonia classification, we also observe that the performance of adversarial models are both less accurate and less stable when comparing with their standard counterpart. One possible explanation is the limited number of pneumonia cases in the whole CheXpert dataset: only 2.4% of CheXpert are pneumonia positive [5], which might restrict the classification performance of adversarial training.

Reviewer 1: Comment 5

Generalisation on robust models section, the model is trained on chexpert and tested on chestx-ray8. Though the adversarial w/ dual bn generalize better than w/o dual bn, the performance of adversarial models is still worse than standard models. This experimental observation should be discussed.

We thank the reviewer for this insightful comment. In Fig. 4 of our manuscript, we evaluate the out of domain generalization of robust classifiers by testing the models on an external chestX-ray8 dataset. Indeed, we observe that a robust model even with dual batch norms still generalizes slightly worse than its standard counterpart when classifying pathologies such as cardiomegaly, edema, and atelectasis. Such an observation is hence added to the manuscript and explained in the corresponding results section:

However, we also observe that a robust model even with dual batch norms still generalizes slightly worse than its standard counterpart when classifying pathologies such as cardiomegaly, edema, and atelectasis. By comparing with Fig. R4, such a behavior is different from the generalization on independent and identically distributed (IID) test sets of the CheXpert data. One contributing factor is the distribution shift between both datasets: although both datasets share similar pathological labels and regions of interests (ROI) on chests, a considerable shift in pixel intensity distribution still exists [6]. Another important factor is the sample complexity of robust learning, which is generally larger than the standard one [7] and therefore always requires more data to generalize (also reflected in Fig. R4).

Fig. R4. *Note that this figure corresponds to the figure 3 in the original manuscript*

More training data and dual batch norms are essential to accurate adversarial training. The classification performance of a standard, an adversarial (Advers.), and an adversarially augmented model (Advers. bns) with respect to different amounts of training data. In accordance with our hypotheses, the performance of adversarially trained models were boosted both by employing the dual batch norm and by enlarging the training set. In the case of pneumonia classification, the performance of adversarially trained models, e.g., Advers. and Advers. bns was limited and less stable due to an insufficient amount of pneumonia positive cases in the dataset.

Reviewer 1: Comment 6

Following above question, the model test performance should also be reported on chexpert dataset, in order to reflect the generalisation gap on new domain.

Thank you for helping us in improving our manuscript. Indeed, the test performance of all models on the chexpert test set should be reported and they can be found in Fig. 2 of the original manuscript and in Fig. R4 which we added to the appendix to guide the intrigued reader.

Reviewer 1: Comment 7

Arguing the visualisation saliency map as a kind of interpretation is somewhat over-claiming. It is more or less a kind of visualised attention map, which is not new actually. Though human-in-the-loop study involves radiologist to evaluate the meaningfulness of such maps, whether these maps indeed help diagnosis, in terms of accuracy or reliability, is unclear and is unlikely to happen in practice.

Thank you very much for your comments which have led us to improve our manuscript. Indeed the usefulness of such saliency maps is a topic of active discussions in the research community. Nevertheless, they have been used in numerous medical studies and have been propagated as a mean to help interpret medical images [8, 9] and continue to be used in more recent studies (e.g. assessing the usefulness of deep learning to diagnose Covid [10]). Thus, even though their usefulness might be debated, they

certainly have established themselves as one of the dominating tools to demonstrate, where the network focuses its attention to. Studies about more detailed investigations of the clinical usability of attention maps are underway or have been done recently [11], but to our knowledge, have been lacking systematic evaluation by multiple radiologists. We felt the need to add this analysis to our study, but we agree with you, that the reader should be guided more closely with regard to the limitations of such saliency maps. We therefore changed our manuscript as follows:

- The subheading **Adversarial Training Selects Clinically Meaningful Features** was changed to **Adversarial Training Selects Features that are close to what Experts consider meaningful**
- In the discussion section, we added the following paragraph: **It should be noted however, that the question of whether these saliency maps indeed help to increase the endpoints of the diagnostic process, e.g. clinical reliability and accuracy, is yet an open research topic and an excellent subject for future work.**

Reviewer 1: Comment 8

Table S3,4,5 should also include the numerical results of robust model w/o batch norms, for clear comparisons.

We again thank the reviewer for this comment. We added the corresponding items for the robust model w/o batch norms to table S3-5 (see below).

Table R2: Comparison of standard and robust models on Luna16 test set.

Prediction	ROC-AUC (95% CI)	p-value	Sensitivity (95% CI)	p-value	Specificity (95% CI)	p-value
Tumor malignancy						
Standard model	0.952 (0.946, 0.959)	-	0.846 (0.827, 0.865)	-	0.933 (0.923, 0.943)	-
Robust model	0.877 (0.866, 0.888)	0.008	0.779 (0.757, 0.800)	0.240	0.810 (0.794, 0.826)	0.065
Robust model, with dual batch norms	0.955 (0.949, 0.961)	0.472	0.893 (0.878, 0.909)	0.308	0.878 (0.864, 0.891)	0.275

Table R3: Comparison of standard and robust models on kneeMRI test set.

Prediction	ROC-AUC (95% CI)	p-value	Sensitivity (95% CI)	p-value	Specificity (95% CI)	p-value
Healthy ACL						
Standard model	0.824 (0.807, 0.841)	-	0.788 (0.773, 0.803)	-	0.730 (0.701, 0.758)	-
Robust model	0.642 (0.622, 0.662)	<0.001	0.697 (0.680, 0.714)	0.256	0.540 (0.508, 0.573)	0.056
Robust model, with dual batch norms	0.825 (0.808, 0.841)	0.487	0.788 (0.773, 0.803)	0.493	0.717 (0.688, 0.745)	0.467
Partially injured ACL						
Standard model	0.742 (0.721, 0.763)	-	0.660 (0.625, 0.696)	-	0.735 (0.719, 0.750)	-
Robust model	0.634 (0.614, 0.655)	0.039	0.536 (0.499, 0.573)	0.213	0.675 (0.658, 0.691)	0.307
Robust model, with dual batch norms	0.741 (0.720, 0.761)	0.482	0.626 (0.590, 0.661)	0.393	0.731 (0.716, 0.747)	0.485
Completely ruptured ACL						
Standard model	0.921 (0.909, 0.933)	-	0.945 (0.915, 0.975)	-	0.802 (0.789, 0.815)	-
Robust model	0.601 (0.557, 0.644)	<0.001	0.610 (0.573, 0.674)	0.031	0.645 (0.629, 0.660)	0.218
Robust model, with dual batch norms	0.918 (0.908, 0.929)	0.499	1.000 (1.000)	0.434	0.798 (0.785, 0.811)	0.494

Table R4: Comparison of standard and robust models on CheXpert test set.

Prediction	ROC-AUC (95% CI)	p-value	Sensitivity (95% CI)	p-value	Specificity (95% CI)	p-value
Cardiomegaly						
Standard model	0.798 (0.782, 0.814)	-	0.727 (0.703, 0.752)	-	0.757 (0.741, 0.774)	-
Robust model	0.826 (0.812, 0.839)	0.348	0.773 (0.750, 0.796)	0.342	0.728 (0.711, 0.745)	0.420
Robust model, with dual batch norms	0.853 (0.841, 0.866)	0.177	0.743 (0.718, 0.767)	0.426	0.809 (0.794, 0.824)	0.334
Edema						
Standard model	0.937 (0.929, 0.946)	-	0.881 (0.859, 0.904)	-	0.881 (0.870, 0.893)	-
Robust model	0.880 (0.868, 0.893)	0.198	0.881 (0.858, 0.904)	0.500	0.744 (0.728, 0.759)	0.161
Robust model, with dual batch norms	0.927 (0.918, 0.936)	0.457	0.928 (0.911, 0.946)	0.401	0.831 (0.818, 0.845)	0.361
Consolidation						
Standard model	0.918 (0.909, 0.927)	-	0.937 (0.918, 0.957)	-	0.806 (0.792, 0.820)	-
Robust model	0.928 (0.910, 0.936)	0.471	0.813 (0.781, 0.844)	0.225	0.865 (0.853, 0.876)	0.355
Robust model, with dual batch norms	0.936 (0.928, 0.943)	0.396	0.906 (0.883, 0.929)	0.465	0.841 (0.828, 0.854)	0.422
Pneumonia						
Standard model	0.857 (0.837, 0.876)	-	1.000 (1.000)	-	0.727 (0.713, 0.741)	-
Robust model	0.810 (0.785, 0.834)	0.373	0.750 (0.681, 0.819)	0.203	0.737 (0.723, 0.751)	0.498
Robust model, with dual batch norms	0.794 (0.768, 0.820)	0.350	1.000 (1.000)	-	0.588 (0.572, 0.604)	0.304
Atelectasis						
Standard model	0.808 (0.794, 0.822)	-	0.787 (0.765, 0.808)	-	0.756 (0.739, 0.773)	-
Robust model	0.751 (0.736, 0.766)	0.161	0.760 (0.737, 0.782)	0.370	0.661 (0.643, 0.680)	0.247
Robust model, with dual batch norms	0.818 (0.804, 0.832)	0.426	0.800 (0.779, 0.821)	0.455	0.772 (0.755, 0.789)	0.454
Pneumothorax						
Standard model	0.865 (0.836, 0.895)	-	0.714 (0.636, 0.792)	-	0.795 (0.782, 0.808)	-
Robust model	0.586 (0.537, 0.635)	0.039	0.572 (0.487, 0.657)	0.369	0.631 (0.615, 0.646)	0.246
Robust model, with dual batch norms	0.791 (0.748, 0.834)	0.330	0.569 (0.484, 0.655)	0.178	0.995 (0.993, 0.997)	0.179
Effusion						
Standard model	0.912 (0.903, 0.921)	-	0.781 (0.758, 0.805)	-	0.870 (0.857, 0.882)	-
Robust model	0.919 (0.910, 0.927)	0.449	0.828 (0.807, 0.849)	0.409	0.870 (0.857, 0.882)	0.500
Robust model, with dual batch norms	0.931 (0.923, 0.939)	0.383	0.797 (0.774, 0.820)	0.429	0.891 (0.879, 0.903)	0.423

Reviewer 1: Comment 9

Real clinical usefulness and practice value of the current work is limited and not sufficiently validated in the paper. These limitations should be discussed.

This comment is well taken and we agree that these limitations should be discussed: There have been too many studies in the past overstating their results with regard to the clinical outlook and we do not want to fall into the same trap. As discussed in Comment 7, we added the following paragraph to the discussion:

It should be noted however, that the question of whether these saliency maps indeed help to increase the endpoints of the diagnostic process, e.g. clinical reliability and accuracy, is yet an open research topic and an excellent subject for future work.

Reviewer 2 Comments

Reviewer 2: Comment 1

The paper evaluates how an adversarial training approach can improve usability of a deep residual convolutional neural network (CNN) architecture for medical image classification. The three key observations of the paper are:

- (1) indeed adversarial attacks can be averted to some degree by this approach;
- (2) dual batch normalisation helps to alleviate the accuracy decrease typically associated with adversarial training;
- (3) with adversarial training the usability of saliency maps marking relevant image areas for classification is improved.

Adversarial attacks exploit slightly "wrinkled" CNN decision boundaries in the image space to identify directions where imperceptible manipulations of the image can lead to misclassifications. The adversarial training strategy employed in this paper anticipates these attacks during training, and adds correspondingly generated correctly labeled images to the training set. This leads to a more robust model, but typically decreases its classification accuracy slightly. Dual batch normalisation can counter this deterioration. The paper shows that adversarial training and dual batch normalisation work as expected on different medical image classification data sets.

We want to thank you for taking the time to review our manuscript and your overall appreciation of our work. We are grateful for the numerous insightful comments that we were happy to address. Please see our specific responses to these comments below.

Reviewer 2: Comment 2

The most interesting and novel observation is that compared to standard training, adversarial training yields saliency maps that are closer to what experts consider relevant image regions. From the results, I don't fully understand if that is indeed an effect of adversarial training, or rather of an increased number of augmented training examples. To clarify this issue one should compare the saliency maps resulting from standard augmentation with e.g., pixel noise with those resulting from adversarial training.

Thank you very much for pointing us to this important aspect. To demonstrate that the interpretability of gradients is closely related to adversarial training, we additionally trained models with random pixel and adversarial noise. We add our experimental results and its explanations to the results section of our manuscript:

Interpretability of gradients is closely related to adversarial training itself and not attributable to a greater number of training images via augmentation. To demonstrate this, additional models were trained by augmenting the input training images with random pixel noise. Fig. R5 visualizes the loss gradient of models trained with medium and strong Gaussian noise augmentation, i.e., $\sigma=0.01$ (1st row) and $\sigma=0.1$ (2nd row), and another model with adversarial training (last row). Based on the rating standard in table 2 of the manuscript, a radiologist assessed the diagnostic relevance of 100 randomly selected chest X-rays and their gradient saliencies. The evaluation scores were 1.14 ± 1.23 , 1.35 ± 1.37 , and 2.14 ± 1.43 for random noise augmentation ($\sigma=0.01$ and $\sigma=0.1$) and adversarial augmentation. Thus we confirmed that the gradients of the loss with respect to the input pixels were semantically meaningful and sparse in the adversarial model, whereas the saliency maps from pixel-noise augmented models were noisy and only loosely connected to occurring pathologies. The sparsity of the gradients can be in parts

explained if we consider the case of generating δ^* via l_∞ bounded fast gradient sign attacks:

$$\begin{aligned}\delta^* \nabla_x \mathcal{L}(x, y) &= \epsilon \text{sign}(\nabla_x \mathcal{L}(x, y)) \nabla_x \mathcal{L}(x, y) \\ &= \epsilon \|\nabla_x \mathcal{L}(x, y)\|_1\end{aligned}\tag{5}$$

Sparser Jacobians in Fig. 6 are obtained via minimizing the loss term above. Several studies have also demonstrated that the high interpretability of gradients is due to adversarial training, because adversarial training leads to confinement of gradients to an image manifold [3, 12].

Fig. R5. Note that this figure corresponds to the figure S4 in the original manuscript. The loss gradient with respect to input pixels of random and adversarial noise augmented models. We additionally trained three models with medium variance Gaussian noise ($\sigma=0.01$), large variance Gaussian noise ($\sigma=0.1$), and adversarial perturbations ($\epsilon=0.005$). As expected, gradients of randomly augmented models are noisy and semantically meaningless, whereas, the adversarially trained model yields interpretable gradient saliencies.

To demonstrate our finding to the interested reader, we included Fig. R5 to the appendix of our manuscript.

Reviewer 2: Comment 3

In the introduction, the authors mention that state of the art deep learning models behave like black-boxes. This is a bit confusing and should be clarified. There are several established approaches that help illuminating their inner workings: CAM (B. Zhou et al. 2016 Learning Deep Features for Discriminative Localization), GradCAM (Selvaraju et al. 2017 Class Activation Mapping GradCAM), guided backpropagation (Springenberg et al. 2014 Striving for Simplicity: The All Convolutional Net), or uncertainty (Gal et al. 2016. "Dropout as a bayesian approximation: Representing model uncertainty in deep learning.") are some examples. The authors should discuss their relationship to the present work, since they use and cite an established technique, to show how adversarial training improves saliency maps.

We agree with the reviewer. Many techniques such as CAM-based [13, 14] or gradient-based [15] methods have been developed to understand the decision making process of deep neural networks. To guide the intrigued reader, we added the following paragraph to the introduction section of our manuscript:

Great effort has been devoted to techniques such as feature and attribution visualization to solve the concern described above. Techniques such as class activation maps, i.e., CAM[13] and GradCAM[14] visualize where the network focusses its attention to. Gradient based methods study, which pixels of the input image are responsible for the neural network firing in a particular way [15]. These methods can be applied irrespective of the way in which the neural network is trained.

Adversarial training offers an efficient way to both counteract adversarial influence and clarify the connection between input and output [16]. Nevertheless, in CV, researchers found that it is generally hard to obtain a both accurate and robust model though adversarial training [17].

Reviewer 2: Comment 4

Please be more specific on the real danger of adversarial attacks in medical imaging. Are there scenarios, or prior results on when and how likely this can happen? This would add to the motivation of the paper.

Thank you very much for this proposition. Since methods of deep learning are currently not yet widespread in clinical practice, we are not aware of an adversarial attack which already had real impact on patient treatment. However, we argue, that in parallel to the introduction of such models, adversaries might use attacking schemes, that exploit the vulnerability of neural networks. In order to guide the intrigued reader, we have added the following paragraph to the introduction section:

To date, adversarial attacks have been of interest primarily to computer science researchers. With the landscape of competing interests within healthcare and with the continuing integration of machine learning in clinical practice, it is likely, that adversaries will arise that exploit such vulnerabilities. To illustrate possible scenarios: insurances could employ deep learning for approval of insurance claims, thus incentivizing adversaries which aim to commit insurance fraud. Another scenario might be a company seeking FDA approval for its newly developed drug in which the efficacy of said drug is measured with a radiological response (e.g. the shrinkage of tumor volume). The company might be tempted to manipulate the follow-up radiological images in an adversarial manner in order to promote its drug's approval [18].

Reviewer 2: Comment 5

The introduction suggests that adversarial training reduces the risk of networks clinging on to so-called "clever hans" strategies, where trivial confounds in the imaging are exploited for classification. It is not clear from the results that this is the case. Please clarify, evaluate, or omit.

Thank you for pointing us to this inaccuracy in our description. We agree, that this point might confuse the reader and does not clearly support or motivate the core message of our paper. We have therefore removed the following section (cited below for your convenience in purple):

The second problem is associated with the so-called 'Clever Hans'-type decision strategy where a model focusses its attention towards irrelevant details such as a hospital department's tag present in the X-ray when assessing the severity of diseases. Thus, patients with X-rays from the intensive care unit might seemingly correctly get classified as pathological even though the model does nothing else than to scan for that specific tag.

The new part replacing this section reads (also see the Comment 4):

To date, adversarial attacks have been of interest primarily to computer science researchers. With the landscape of competing interests within healthcare and with the continuing integration of machine learning in clinical practice, it is likely, that adversaries will arise that exploit such vulnerabilities. To

illustrate possible scenarios: insurances could employ deep learning for approval of insurance claims, thus incentivizing adversaries which aim to commit insurance fraud. Another scenario might be a company seeking FDA approval for its newly developed drug in which the efficacy of said drug is measured with a radiological response (e.g. the shrinkage of tumor volume). The company might be tempted to manipulate the follow-up radiological images in an adversarial manner in order to promote its drug’s approval [18].

Reviewer 2: Comment 6

In the results section, the use of AUC-ROC is helpful but slightly limited for unbalanced data sets, such as pathology detection. Precision-recall analysis offers a more informative assessment of the classification results in these scenarios.

We appreciate your help in improving our manuscript. As suggested by you, we added a precision-recall analysis to offer the reader a more informative assessment of the classification results. Fig. R6 visualizes the precision-recall values in our models on the CheXpert, Rijeka, and Luna datasets. We included Fig. R6 in the appendix of our manuscript. The following sentences are added to the results section:

To select the best model in the task of pathology detection, we quantified the model’s performance via the area under the receiver operating characteristic curve (ROC-AUC) and precision-recall curve.

Reviewer 2: Comment 7

Dual batch norm strategy is good and has been recently described as an approach to counter accuracy degradation due to adversarial learning, e.g., in Wang et al. NIPS 2020, "Once-for-All Adversarial Training: In-Situ Tradeoff between Robustness and Accuracy for Free" Xie et al. CVPR 2020 "Adversarial examples improve image recognition" Xie et al. ICLR 2020. "Intriguing properties of adversarial training at scale" (already cited by the authors, in its arxiv version) These paper should be cited, and the relationship should be commented on.

We thank the reviewer for bringing the above publications to our attention. Similar to our study, the model settings of dual batch norms have been used in the mentioned works of Xie et al. [1, 2] and Wang et al. [19]. Particularly, Xie at al. found that batch norm layers may negatively affect the adversarial robustness and proposed to add another branch of batch norm layers to account for the different statistics (running mean and variance) between clean and adversarially perturbed batches [2]. In their later work [1], Xie at al. improved the classification accuracy on ImageNet by using adversarial augmented training and dual batch normalizations. Also motivated by the difference between feature statistics of standard and adversarially perturbed samples, Wang et al., select a hyper parameter λ to balance the trade-off between robustness and accuracy [19]. However, in the present work, we offer another explanation to the usage of dual batch norms through the lens of loss landscape (see section Revisiting adversarially augmented training) and learned feature similarity (see section Comment 8).

We added the following sentence to the discussion section:

When applied to ImageNet classification tasks, the authors of [1] achieved a top-1 accuracy increase when performing adversarially augmented training. This finding is not reflected by our study, which matched, but not increased accuracy of the non-adversarially trained models. A potential reason for this might be the difference in dataset size and problem complexity: medical images are both rarer as a whole and greater in terms of pixel numbers individually, making the problem considerably harder.

Fig. R6. Note that this figure corresponds to the figure S1 in the original manuscript. Precision-recall analysis also suggests that the setting of dual batch norms is beneficial to improve the performance of robust models. We compared the classification performance of three models, namely, standard, adversarially trained, and adversarially trained with dual batch norms, against the test sets of the ground truth in the precision-recall space. Note that the adversarially trained model performs comparable to the standard model only when employing dual batch norms.

However, the results of [2, 1, 19] and our results in the experiments with reduced data indicate, that accuracy of adversarially trained models with separate batch norms might surpass the standard networks if sufficient amount of data is available.

Reviewer 2: Comment 8

The correlation between layers (can it be viewed as a collapse?) observed in models resulting from adversarial training, and the resolution of this correlation by dual batch training is a very interesting and insightful observation. Could you please discuss how this relates to prior work on adversarial training and dual batch normalisation.

We thank the reviewer for this positive comment.

The high correlation between features learned by robust layers is reflected by the overall high CKA

value in Fig. 7 B and D of the manuscript. The linear CKA used in our study is defined via:

$$\begin{aligned} \text{CKA}(X, Y) &= \text{CKA}(XX^T, YY^T) \\ &= \frac{\|Y^T X\|_F^2}{\|X^T X\|_F \|Y^T Y\|_F} \end{aligned} \quad (6)$$

Considering eigen-decomposition, we let λ_X^i and \mathbf{u}_X^i be the i^{th} eigenvalue and eigenvector of the dot products between the representations of the samples XX^T . The linear CKA can be rewritten as [20]:

$$\text{CKA}(X, Y) = \frac{\sum_i \sum_j \lambda_X^i \lambda_Y^j}{\sqrt{\sum_i (\lambda_X^i)^2} \sqrt{\sum_j (\lambda_Y^j)^2}} \langle \mathbf{u}_X^i, \mathbf{u}_Y^j \rangle^2. \quad (7)$$

The term $\langle \mathbf{u}_X^i, \mathbf{u}_Y^j \rangle$ reveals that the overall high CKAs of a robust model is a result of feature eigenvectors of adversaries trained layers that tend to align to a similar direction. Via the lens of dual batch norms (Fig. 7 C and D), the link between batch norms and the model accuracy is reflected by the inner product change of feature eigenvectors when switching between batch norm branches, i.e., BN_{std} and BN_{adv} .

To find the link between feature similarity and robust learning, a previous work by Zhang, H. et al. [21] improved model robustness against l_∞ attacks via optimizing similarities between features in the minibatch. Using the proposed feature-based adversarial training, they obtain a more regularized loss landscape and avoided label information leakage. However, to the best of our knowledge, no previous study compared the interlayer feature dependencies between standard and robust models. The importance of batch norms in adversarial vulnerability was studied in [22]. Instead of using simple models [22], our ResNet-50 model with dual batch norms offers the opportunity to inspect the influence of batch norms on robustness (loss landscape in Fig. R2) and accuracy (feature similarity in Fig. 7).

To provide the interested reader with our explanations we have added the above paragraph under "Linear CKA and Relation to Prior Work" to the appendix.

Reviewer 2: Comment 9

Evaluating the saliency maps by expert assessment is a good experiment.

Thank you very much for your appreciation.

Reviewer 2: Comment 10

Please evaluate if the improvement of the saliency maps is due to an increased number of augmented training examples, that would be also achieved if augmentation was done with pixel noise, or if it is truly linked to the adversarial strategy. This is not clear. Please also compare with other state of the art approaches that high-light relevant image areas such as e.g., GradCAM.

Thank you very much for pointing us to this shortcoming of our analysis. We agree that such an analysis had been missing so far and have performed the following experiment: Next to the adversarial training as described in the original version of the manuscript, we augmented the standard training with noise augmented samples, such that the number of augmented images in the adversarial setting matched that of the noise-augmented training. We added gaussian pixel noise of two different strengths ($\sigma = 0.01$

and $\sigma = 0.1$) and visualized the gradient saliency maps of the such trained networks next to the gradient saliency maps of the adversarially trained images in Fig. R5. As visualized in Fig. R5, the saliency maps after adversarial training with dual batch norms were more closely related to the pathology. This was also confirmed by an additional quantitative analysis by one radiologist with 8 years of experience: Based on the rating standard in table 2 of the manuscript, we assessed the diagnostic relevance of 100 randomly selected chest X-rays and their gradient saliencies by a radiologist. The evaluation scores were 1.14 ± 1.23 , 1.49 ± 1.37 , and 2.14 ± 1.43 for random noise augmentation ($\sigma=0.01$ and $\sigma=0.1$) and adversarial augmentation.

We also investigated the effect of adversarial training on GradCAM, see Fig. R7. GradCAM is a technique with limited resolution: it points to the approximate image region, but fails to exactly delineate smaller image regions responsible for the diagnosis. Hence, the difference between adversarially trained networks with dual batch norms and standard networks was less pronounced and the diagnostic rating was lower in general: Based on the rating standard in table 2 of the manuscript, we assessed the diagnostic relevance of 100 randomly selected chest X-rays and their gradient saliencies by one radiologist. The evaluation scores were 0.83 ± 0.67 and 0.98 ± 0.72 for the GradCAM with standard training and the GradCAM with dual batch norm adversarial training respectively.

To provide these findings to the research community we added Fig. R7 and the following paragraph to the appendix:

GradCAM Visualization

We also investigated the effect of adversarial training on GradCAM, see Fig. R7. GradCAM is a technique with limited resolution: it points to the approximate image region, but fails to exactly delineate smaller image regions responsible for the diagnosis. Hence, the difference between adversarially trained networks with dual batch norms and standard networks was less pronounced and the diagnostic rating was lower in general: Based on the rating standard in table 2 of the manuscript, we assessed the diagnostic relevance of 100 randomly selected chest X-rays and their gradient saliencies by one radiologist. The evaluation scores were 0.83 ± 0.67 and 0.98 ± 0.72 for the GradCAM with standard training and the GradCAM with dual batch norm adversarial training respectively.

Fig. R7. Note that this figure corresponds to the figure S5 in the original manuscript. The robust model locate abnormalities in radiographs using GradCAM. Both standard (2nd column) and adversarially trained (3rd column) models are able to localize abnormalities in chest radiographs. Patients from CheXpert test set with pleural effusion, cardiomegaly, and edema are listed in 1-3 rows, respectively. GradCAMs are generated by the most confident correct class of the model.

References

- [1] C. Xie, M. Tan, B. Gong, J. Wang, A. L. Yuille, and Q. V. Le, “Adversarial examples improve image recognition,” in *Proceedings of the IEEE/CVF Conference on Computer Vision and Pattern Recognition*, pp. 819–828, 2020.
- [2] C. Xie and A. Yuille, “Intriguing properties of adversarial training at scale,” *arXiv preprint arXiv:1906.03787*, 2019.
- [3] A. Ilyas, S. Santurkar, D. Tsipras, L. Engstrom, B. Tran, and A. Madry, “Adversarial examples are not bugs, they are features,” *arXiv preprint arXiv:1905.02175*, 2019.
- [4] S. Santurkar, D. Tsipras, A. Ilyas, and A. Madry, “How does batch normalization help optimization?,” *arXiv preprint arXiv:1805.11604*, 2018.
- [5] J. Irvin, P. Rajpurkar, M. Ko, Y. Yu, S. Ciurea-Ilcus, C. Chute, H. Marklund, B. Haghgoo, R. Ball, K. Shpanskaya, *et al.*, “Chexpert: A large chest radiograph dataset with uncertainty labels and

- expert comparison,” in *Proceedings of the AAAI Conference on Artificial Intelligence*, vol. 33, pp. 590–597, 2019.
- [6] T. Han, S. Nebelung, C. Haarburger, N. Horst, S. Reinartz, D. Merhof, F. Kiessling, V. Schulz, and D. Truhn, “Breaking medical data sharing boundaries by using synthesized radiographs,” *Science Advances*, vol. 6, no. 49, p. eabb7973, 2020.
- [7] L. Schmidt, S. Santurkar, D. Tsipras, K. Talwar, and A. Madry, “Adversarially robust generalization requires more data,” *arXiv preprint arXiv:1804.11285*, 2018.
- [8] A. Mitani, A. Huang, S. Venugopalan, G. S. Corrado, L. Peng, D. R. Webster, N. Hammel, Y. Liu, and A. V. Varadarajan, “Detection of anaemia from retinal fundus images via deep learning,” *Nature biomedical engineering*, vol. 4, no. 1, pp. 18–27, 2020.
- [9] N. Bien, P. Rajpurkar, R. L. Ball, J. Irvin, A. Park, E. Jones, M. Bereket, B. N. Patel, K. W. Yeom, K. Shpanskaya, *et al.*, “Deep-learning-assisted diagnosis for knee magnetic resonance imaging: development and retrospective validation of mrnet,” *PLoS medicine*, vol. 15, no. 11, p. e1002699, 2018.
- [10] A. Mangal, S. Kalia, H. Rajgopal, K. Rangarajan, V. Namboodiri, S. Banerjee, and C. Arora, “Covidaid: Covid-19 detection using chest x-ray,” *arXiv preprint arXiv:2004.09803*, 2020.
- [11] N. T. Arun, N. Gaw, P. Singh, K. Chang, K. V. Hoebel, J. Patel, M. Gidwani, and J. Kalpathy-Cramer, “Assessing the validity of saliency maps for abnormality localization in medical imaging,” *arXiv preprint arXiv:2006.00063*, 2020.
- [12] B. Kim, J. Seo, and T. Jeon, “Bridging adversarial robustness and gradient interpretability,” *arXiv preprint arXiv:1903.11626*, 2019.
- [13] B. Zhou, A. Khosla, A. Lapedriza, A. Oliva, and A. Torralba, “Learning deep features for discriminative localization,” in *Proceedings of the IEEE conference on computer vision and pattern recognition*, pp. 2921–2929, 2016.
- [14] R. R. Selvaraju, M. Cogswell, A. Das, R. Vedantam, D. Parikh, and D. Batra, “Grad-cam: Visual explanations from deep networks via gradient-based localization,” in *Proceedings of the IEEE international conference on computer vision*, pp. 618–626, 2017.
- [15] J. T. Springenberg, A. Dosovitskiy, T. Brox, and M. Riedmiller, “Striving for simplicity: The all convolutional net,” *arXiv preprint arXiv:1412.6806*, 2014.
- [16] A. Madry, A. Makelov, L. Schmidt, D. Tsipras, and A. Vladu, “Towards deep learning models resistant to adversarial attacks,” *arXiv preprint arXiv:1706.06083*, 2017.
- [17] D. Tsipras, S. Santurkar, L. Engstrom, A. Turner, and A. Madry, “Robustness may be at odds with accuracy,” *arXiv preprint arXiv:1805.12152*, 2018.
- [18] S. G. Finlayson, J. D. Bowers, J. Ito, J. L. Zittrain, A. L. Beam, and I. S. Kohane, “Adversarial attacks on medical machine learning,” *Science*, vol. 363, no. 6433, pp. 1287–1289, 2019.
- [19] H. Wang, T. Chen, S. Gui, T.-K. Hu, J. Liu, and Z. Wang, “Once-for-all adversarial training: In-situ tradeoff between robustness and accuracy for free,” *arXiv preprint arXiv:2010.11828*, 2020.
- [20] S. Kornblith, M. Norouzi, H. Lee, and G. Hinton, “Similarity of neural network representations revisited,” in *International Conference on Machine Learning*, pp. 3519–3529, PMLR, 2019.
- [21] H. Zhang and J. Wang, “Defense against adversarial attacks using feature scattering-based adversarial training,” *arXiv preprint arXiv:1907.10764*, 2019.
- [22] A. Galloway, A. Golubeva, T. Tanay, M. Moussa, and G. W. Taylor, “Batch normalization is a cause of adversarial vulnerability,” *arXiv preprint arXiv:1905.02161*, 2019.

Reviewers' Comments:

Reviewer #1:

Remarks to the Author:

The authors have taken the previous comments carefully, and provided satisfactory response to all the questions. The contents are well integrated into the manuscript. I have no further comments.

Reviewer #2:

Remarks to the Author:

In the revised version of the manuscript, the authors have addressed the points raised by the reviewers. The clarifications made in the introduction regarding the claims of the paper, the additional experimental evaluation that illustrates fine points are helpful and add value to the manuscript.

The text added to the manuscript regarding Rev2-Comment4 is helpful, and more clear now. A slight revision to use the last paragraph of the introduction to more specifically point at both the empirical and theoretical contributions of the paper would be good. Especially the link between interpretability and the reshaping of the cost function as illustrated in the figures S1/2 seem central to the paper and that should be made clear.

Overall - since a lot of information has been added in the revision - the manuscript would benefit from a parse to consolidate the key claims in the introduction, and the corresponding experimental results. Structuring the result sections so that the key insights of every results are easier to discern from additional detail information would be helpful.

Minor point:

Typo in the manuscript text referred to in the answer to Rev2-Comment3: adversarial -> adversarial.

Response to the reviewers

Reviewer 1 Comments

Reviewer 1: Comment 1

The authors have taken the previous comments carefully, and provided satisfactory response to all the questions. The contents are well integrated into the manuscript. I have no further comments.

Thank you very much for your guidance and advice in improving our manuscript. We are happy to have satisfactorily addressed all of the issues raised by you.

Reviewer 2 Comments

Reviewer 2: Comment 1

In the revised version of the manuscript, the authors have addressed the points raised by the reviewers. The clarifications made in the introduction regarding the claims of the paper, the additional experimental evaluation that illustrates fine points are helpful and add value to the manuscript.

We again thank our 2nd reviewer for taking the time to review our manuscript and your overall appreciation of our last revision.

Reviewer 2: Comment 2

The text added to the manuscript regarding Rev2-Comment4 is helpful, and more clear now. A slight revision to use the last paragraph of the introduction to more specifically point at both the empirical and theoretical contributions of the paper would be good. Especially the link between interpretability and the reshaping of the cost function as illustrated in the figures S1/2 seem central to the paper and that should be made clear.

We thank the reviewer for this helpful suggestion. To address the suggestion of this point, we mainly modify the last part of our introduction section and highlight the gradient interpretability and the observed reshaping of the loss surface. This part now reads:

In this paper, we address the mentioned degradation of accuracy via investigating loss landscapes of robust and non-robust models. More importantly, we find that training the model in this fashion allows for the model's reasoning to be more closely aligned with clinical expectations than when the model is trained in a conventional fashion.

Reviewer 2: Comment 3

Overall - since a lot of information has been added in the revision - the manuscript would benefit from a parse to consolidate the key claims in the introduction, and the corresponding experimental results. Structuring the result sections so that the key insights of every results are easier to discern from additional detail information would be helpful.

Indeed, we agree with the reviewer that we should restructure the subsections in our results section in order to clearly deliver our main message.

Now, we reorganized the subsections into four main sections:

- Adversarial Training Smoothes the Loss Surface
- Revisiting Adversarially Augmented Training
- Adversarial Training Selects Features that are Close to What Experts Consider Meaningful
- Batch Normalization Influences Learned Representations

Reviewer 2: Comment 4

Minor point: Typo in the manuscript text referred to in the answer to Rev2-Comment3: adversarial → adversarial

We have corrected it in our manuscript accordingly.